# Fabrication and Optimization of Electrospun Shellac Fibers Loaded with *Senna alata* Leaf Extract

**DOI:** 10.3390/polym16020183

**Published:** 2024-01-08

**Authors:** Wah Wah Aung, Wantanwa Krongrawa, Sontaya Limmatvapirat, Pattranit Kulpicheswanich, Siriporn Okonogi, Chutima Limmatvapirat

**Affiliations:** 1Department of Industrial Pharmacy, Faculty of Pharmacy, Silpakorn University, Nakhon Pathom 73000, Thailand; wahwahaung31@gmail.com (W.W.A.); krongrawa_w@su.ac.th (W.K.); limmatvapirat_s@su.ac.th (S.L.); 2Pharmaceutical Biopolymer Group (PBiG), Faculty of Pharmacy, Silpakorn University, Nakhon Pathom 73000, Thailand; 3Baan Teraeng Groups Co., Ltd., 12/32 Moo 19 Chumhed, Muang Buriram, Buriram 31000, Thailand; teraeng007@gmail.com; 4Center of Excellence in Pharmaceutical Nanotechnology, Faculty of Pharmacy, Chiang Mai University, Chiang Mai 50200, Thailand; siriporn.okonogi@cmu.ac.th; 5Department of Pharmaceutical Sciences, Faculty of Pharmacy, Chiang Mai University, Chiang Mai 50200, Thailand

**Keywords:** electrospun fibers, fractional factorial design, Box–Behnken design, shellac, *Senna alata*, rhein, antimicrobial activity

## Abstract

Single-fluid electrospinning creates nanofibers from molten polymer solutions with active ingredients. This study utilized a combination of a fractional factorial design and a Box–Behnken design to examine crucial factors among a multitude of parameters and to optimize the electrospinning conditions that impact fiber mats’ morphology and the entrapment efficiency of *Senna alata* leaf extract. The findings indicated that the shellac content had the greatest impact on both fiber diameter and bead formation. The optimum electrospinning conditions were identified as a voltage of 24 kV, a solution feed rate of 0.8 mL/h, and a shellac–extract ratio of 38.5:3.8. These conditions produced nanosized fibers with a diameter of 306 nm, a low bead-to-fiber ratio of 0.29, and an extract entrapment efficiency of 96% within the fibers. The biphasic profile of the optimized nanofibers was confirmed with an in vitro release study. This profile consisted of an initial burst release of 88% within the first hour, which was succeeded by a sustained release pattern surpassing 90% for the next 12 h, as predicted with zero-order release kinetics. The optimized nanofibers demonstrated antimicrobial efficacy against diverse pathogens, suggesting promising applications in wound dressings and protective textiles.

## 1. Introduction

An additional name for candle bush is *Senna alata* (L.) Roxb. (SA), which belongs to the Fabaceae family. This ornamental plant originates in its natural habitat, the Amazon Rainforest. It is widely accessible across the continents of Asia and Africa. Cultivated for medicinal purposes in the Philippines, Thailand, and Indonesia, this shrub is extensively distributed [1]. As a result of its ethnomedical applications in the treatment of numerous health conditions, it has acquired considerable significance. Predominantly found in active phytochemicals [1], compared with the roots, flowers, and stem materials, the leaves are utilized more frequently in traditional medicine. SA leaves, both fresh and dried, have been utilized as remedies for skin maladies, constipation, stomach pain, and ringworm in a number of countries [2]. SA leaf extract has been associated with a range of pharmacological properties [3,4,5,6,7], which include analgesic, laxative, anti-inflammatory, antioxidant, anthelmintic, and antimicrobial effects. SA leaf extract is rich in carbohydrates, proteins, anthraquinones, flavonoids, terpenoids, tannins, phlobatannins, saponins, cardiac glycosides, and flavonoids [8]. The authors, Ahmed S. and Shohael A.M., identified antifungal anthraquinones, namely, aloe-emodin, chrysophanol, emodin, and rhein, in SA leaves [9]. Khare C.P. noted in *Indian Medicinal Plants* that rhein was responsible for the antibacterial activity of SA leaves [10]. Limmatvapirat, C., et al. discovered that the high rhein content in SA leaf extract was responsible for its antioxidant, anti-inflammatory, and antibacterial properties [11]. This demonstrated that SA leaf extract possessed sufficient biological properties to be utilized in the formulation of pharmaceuticals.

Shellac, which originates from the resinous secretions of lac insects (*Laccifer lacca*), is a naturally occurring polymer [12]. Single esters of aleuritic acid and terpenic acids (specifically, jalaric acid or laccijalaric acid) comprise its composition. Shellac, which has a low crystallinity and a melting point between 50 °C and 75 °C, is a semicrystalline polymer [13]. Shellac is applicable in numerous industries, including food and pharmaceutical technologies, due to its numerous advantageous properties. The aforementioned characteristics consist of favorable film formation, water resistance, flexibility, non-toxicity, biodegradability, high luster, acid resistance, and low gas and water vapor permeability [14]. Additionally, delivery systems composed of shellac and varying in size from nanometers to micrometers have been developed [15]. Examples include nanofibers, microparticles, hydrogels, and others.

Sticklac is a resinous substance that is secreted by female lac insects, a species that inhabits tree branches. Sticklac was subjected to a series of procedures in order to obtain seedlac: crushing, sieving, washing with an alkaline aqueous solution and water, and air-drying to eliminate insect remains, branches, and other impurities [16]. When seedlac is purified, the valuable substance shellac is produced. There are two distinct methods for producing shellac: melting, filtration, and forming into thin sheets; or dissolving the seedlac in an organic solvent, filtration to remove sediment, evaporation, and establishing into a thin film [16]. The three distinct varieties of commercial shellac correspond to the method of production: bleached, machine-made, and handmade. By dissolving seedlac in an alkaline aqueous solution, treating it with sodium hypochlorite, and precipitating it with sulfuric acid, bleached shellac is produced. By removing wax from the precipitate via filtration, dewaxed or bleached shellac is produced [17]. Although several studies have been published on the application of bleached shellac for enteric coating of drugs [18], edible surface coating of fruits and vegetables [19], microparticle formation of antibacterial oils [20], and gel formation for periodontitis treatment [21], there is currently no research examining the application of bleached shellac in the production of electrospun nanofibers.

Drug delivery systems based on nanotechnology use nanoscale materials to regulate and target drug release with extreme precision. Nanotechnology enables the site-specific and target-oriented medication delivery that is beneficial for chronic diseases. The application of natural products integrated into nanomaterials in clinical settings remains a challenging endeavor [22]. A range of biopolymeric materials are used in drug delivery and natural product-based nanotechnology, such as shellac [23], chitosan [24], xanthan gum [24], and cellulose [25]. At present, a multitude of scientific investigations are focused on the pharmacological potential of bioactive compounds or phytochemical constituents found in various plant species. The objective is to create active ingredients that exhibit superior safety and reduced adverse effects compared with synthetic compounds currently in use [26].

Due to their numerous advantages, nanofibers, which are defined as fibers with a diameter of less than 1000 nm, are extensively utilized in drug delivery, cosmetics, and wound dressing. The advantages encompass favorable stability, accurate administration of active ingredients to the intended site, minimal toxicity, enhanced capacity for loading drugs, exceptional mechanical properties, encapsulation of various categories of active compounds, and suitability for pharmaceuticals that are sensitive to temperature [27]. Electrospinning (including emulsion electrospinning, coaxial electrospinning, and multi-jet electrospinning) and non-electrospinning processes (including interfacial polymerization, phase separation, and self-assembly) are utilized to generate nanofibers [28]. Electrospinning is the predominant and most efficient method utilized in the fabrication of nanofibers. Single-fluid electrospinning involves the feeding of a polymer solution or polymer melt containing active components through a high-voltage electrospinning machine (10–50 kV). This process yields delicate fibers [29]. Applied voltage, polymer type, and polymer concentration frequently have a significant impact on the properties of nanofibers [30]. The electrospinning process utilizes electrostatic force to deform electrospinning fluid into nanoscale fibers that resemble the extracellular matrix of native tissue. As a result, these filaments have the capacity to facilitate regular cellular processes, such as cell proliferation and attachment. Electrospinning yields nanofibers that demonstrate remarkable characteristics, such as a substantial specific surface area, elevated porosity, favorable biocompatibility, and biodegradability. Consequently, electrospun nanofibers have been extensively implemented across various sectors of the pharmaceutical industry, including but not limited to wound dressings, medical implants, drug delivery systems, and scaffolds for tissue engineering [31].

The physical and chemical properties of electrospun nanofibers are primarily dictated by the electrospinning process, with a strong reliance on the morphology of these nanofibers. The electrospinning process is notably influenced by three categories of factors: process parameters, solution parameters, and ambient conditions [32]. Modifying these parameters has the potential to bring about alterations in the morphology and structure of the nanofibers. Conducting an analysis of the optimal electrospinning conditions becomes imperative for achieving the desired array of fine-diameter, beadless nanofibers. A fractional factorial design (FFD) stands out as a crucial statistical advancement for investigating the impacts of multiple controllable factors on a specific response of interest. The careful reduction in the size of an experiment is not only a well-established design feature but also serves to prevent the loss of valuable data, given that not all conceivable combinations of the levels of the factors of interest are tested [33]. This design functions as a preliminary stage in the assessment procedure, wherein critical factors are identified prior to advancing to a more exhaustive evaluation. Reducing the number of trials does not encompass all potential interactions that may occur among independent factors within the experimental space. After conducting preliminary tests, crucial variables are identified for additional analysis with the utilization of either a response surface design or a full factorial design, which includes a central tendency and replicated experiments. In contrast to three-level factorial designs, response surface designs present a more streamlined methodology by generating significant insights with a reduced number of trials. In order to develop a more accurate representation of the response variable, these designs examine curvature and interactions in the experimental space [34,35,36]. Hence, utilizing an alternative optimization design becomes crucial for achieving optimal conditions. One frequently used response surface design is the Box–Behnken (BB) design, which aims to optimize processes by examining the interplay between various factors and their impact on a response variable. The BB design, known for its rotatability, requires three levels per factor and incorporates a central point for quadratic effect estimation. This approach is particularly well-suited for experiments involving more than two factors, providing a streamlined method to determine optimal conditions with minimal iterations [37]. Once the response surface model is established, optimization techniques can be applied to identify optimal conditions for maximizing or minimizing the response variable.

Due to their unique characteristics, electrospun nanofiber dressings promote more effective wound healing than conventional dressings. Nanofibers can optimally absorb exudates, provide a moist environment that promotes cell respiration and proliferation, reduce bacterial infection and inflammation, offer additional high permeability, and protect damaged tissues from dehydration [38]. Electrospinning can incorporate numerous molecules into nanofibers, such as drugs, growth factors, bioactive nanoparticles, and herbal extracts. Because of their biodegradability, high compatibility with blood and tissues, and antimicrobial and anti-inflammatory properties, electrospun biopolymer dressings containing herbal extract or natural compounds could promote wound healing and cell growth [39,40,41]. Sprague Dawley rats subjected to treatment with electrospun gelatin membranes containing *Centella asiatica* extract as transdermal wound dressings demonstrated superior dermal wound-healing activity in comparison with rats treated with gauze (utilized as a control), gelatin membranes without *C. asiatica* extract, and commercial wound dressings. This enhanced activity can be attributed to the antibacterial and anti-inflammatory properties of the electrospun membranes [42]. The wound dressing, composed of chitosan/polyethylene oxide nanofibers containing green tea (*Camellia sinensis*) extract and prepared with the electrospinning method, exhibited superior healing effects on rat wounds compared with other prepared wound dressings. This notable effectiveness can be attributed to its antibacterial activities against *Escherichia coli* and *Staphylococcus aureus*, as well as its high swelling capacity. This characteristic ensures the retention of moisture on the wound surface throughout the healing process and prevents nanofibers from adhering to the wound surface, contributing to an optimal healing environment [43]. Electrospun gelatin nanofibers containing an ethanol extract of *Curcuma comosa* Roxb. rhizomes displayed antioxidant, anti-tyrosinase, and antibacterial properties. To achieve optimal conditions for electrospun nanofibers with enhanced freeze–thaw stability, the gelatin concentration was 30% *w*/*v* in a co-solvent system of acetic acid and water (9:1 *v*/*v*) at a feed rate of 3 mL/h and an applied voltage of 15 kV. The lowest loading percentage of 5% *w*/*v* ethanol extract in nanofibers showed remarkable DPPH radical scavenging, anti-tyrosinase, and antibacterial properties against *S. aureus* and *Staphylococcus epidermidis* [44]. Although numerous investigations [45,46,47,48,49] have detected noteworthy antimicrobial capabilities of SA leaf extract, no research has yet been conducted on its incorporation into electrospun nanofibers. Based on the antimicrobial properties of SA leaf extract and the unique characteristics of electrospun shellac fibers, it is possible that electrospun shellac fibers infused with this extract could promote faster wound healing.

The present study used electrospinning to generate shellac fibers that incorporated SA leaf extract. The screening procedure was used to determine the most effective electrospinning parameters that would yield nanofibers with the intended morphology, utilizing a fractional factorial experimental design. In order to determine the optimal values for each dependent parameter, the BBD was applied. Furthermore, an investigation was conducted on the entrapment efficiency and release kinetics of rhein from electrospun shellac fibers laden with SA leaf extract. Additionally, the antimicrobial properties of these fibers were assessed, given that inhibiting microbial growth could accelerate the healing of wounds.

## 2. Materials and Methods

### 2.1. Materials

SA leaves, bleached shellac, and ethanol absolute (≥99.8% *v*/*v*) were procured from Charoensuk Osot, an herbal shop in Nakhon Pathom, Thailand; Excelacs Co., Ltd., Bangkok, Thailand; and VWR International (Fontenay-sous-Bois, Val-de-Marne, France), respectively. Acetonitrile HPLC grade (≥99.93% *v*/*v*), orthophosphoric acid (85% *w*/*w*), and standard rhein (≥95% *w*/*w*) were procured from Fisher Scientific Korea Ltd. (Gangnam-gu, Seoul, Korea), Ajax Finechem (Botany, Auckland, New Zealand), and MilliporeSigma Supelco (Frankfurter Strasse, Darmstadt, Germany). Tryptic Soy Broth (TSB) and Tryptic Soy Agar (TSA) were purchased, respectively, from Becton, Dickinson and Company (Sparks, MD, USA) and HiMedia Laboratories Private Limited (Mumbai, Maharashtra, India).

### 2.2. Preparation and Phytochemical Analysis of SA Leaf Extract

In accordance with our prior research [11], SA leaf extraction was conducted under optimal extraction conditions in order to yield an extract with a high rhein content. A precise volume of dehydrated pulverized SA leaves was combined with 95% *v*/*v* ethanol in a 25:1 (mL/g) solid-to-solvent ratio. The solution was subsequently extracted utilizing an ultrasonicator (Model 230D, Crest Ultrasonics Corporation, Trenton, NJ, USA) set to a frequency of 42–45 kHz (level 9) at a temperature of 60 °C. Following an extraction period of 18 min, the obtained solutions underwent filtration through Whatman filter paper No. 1. Subsequently, the filtrate was evaporated using a rotary evaporator (R-100, Buchi, Tokyo, Japan) set at 40 °C and 35 mbar of pressure. The concentrated mass was thoroughly dried using a freeze dryer (Model 6112974, Labconco Corporation, Kansas City, MO, USA) and subsequently stored in a dark environment at −20 °C until further analysis.

Phytochemical analysis of SA leaf extract, as detailed by Karthika, C. et al. [8], was conducted using modified methods. The identification of phytochemical constituents relied on characteristic color changes and precipitation reactions following standard procedures. Anthraquinones were discerned by partitioning the HCl hydrolyzed extract with benzene, resulting in a red color in the separated benzene layer upon the addition of NH_4_OH. Flavonoids were indicated by the red color of the extract solution after the addition of concentrated HCl. Terpenoids were identified by the formation of a reddish-brown color in the CHCl_3_ layer when the extract solution was mixed with CHCl_3_ and slowly treated with concentrated H_2_SO_4_. Alkaloids produced a red-orange precipitate after the extract was dissolved in a 5% *v*/*v* HCl solution and treated with Dragendorff reagent. Saponins were recognized by the persistent foam produced during shaking with hot water for 5 min. Tannins were confirmed by the blue-black coloration of the extract solution upon the addition of FeCl_3_ reagent. Lastly, steroids were detected by spotting the extract solution on a silica gel F254 thin-layer chromatography (TLC) plate, developing it in a mobile phase of CH_2_Cl_2_:MeOH (4:1, *v*/*v*), and observing a black spot after spraying the TLC plate with 10% *v*/*v* H_2_SO_4_ in MeOH before heating.

The examination of rhein was performed on a silica gel F254 TLC plate using a mobile phase consisting of EtOAc:MeOH:H_2_O (100:17:13, *v*/*v*/*v*). Following the air-drying of the TLC plate, the presence of a rhein spot was revealed under UV light at a wavelength of 254 nm.

### 2.3. Design of Experiments

#### 2.3.1. Fractional Factorial Design (FFD)

An FFD was used to investigate the main effects and potential interactions among various factors influencing multiple response variables in shellac fibers incorporated with SA leaf extract. The parameters and experimental ranges, as outlined in Table 1, were established with preliminary experiments and prior research with slight modifications [23,50]. Design Expert software, version 8.0.6 (Stat-Ease Inc., Minneapolis, MN, USA), was utilized to create the design matrix and perform data analysis. The response variables measured were the diameter of fibers and the bead-to-fiber ratio.

#### 2.3.2. Box–Behnken Design (BBD)

The FFD results aid in the identification of critical variables that have a substantial impact on both the diameter of the fiber and the ratio of beads to fibers. In order to optimize the electrospinning procedure for shellac nanofibers laden with SA leaf extract, a BBD was executed. Multiple response variables, including fiber diameter, the bead-to-fiber ratio, and entrapment efficacy, were accounted for in this design. The precise ranges and levels of each parameter are specified in Table 2. A total of 29 experimental trials were conducted, with the inclusion of five center points in order to guarantee thorough coverage of the data.

### 2.4. Preparation and Evaluation of Shellac-SA Leaf Extract Solutions

In order to formulate the solution, an exact volume of SA leaf extract was dissolved in ethanol with a concentration of 95% *v*/*v*. To guarantee total solubility, an ultrasonicator (Model 230D, Crest Ultrasonics Corporation, Trenton, NJ, USA) was used. Following that, different concentrations of bleached shellac were introduced into each of these solutions and agitated for one night at ambient temperature using a magnetic stirrer (ST10, Finepcr, Seoul, Republic of Korea). After agitation, the concentrations of each solution were modified by adding 95% *v*/*v* ethanol in order to achieve the intended results. Following that, an examination was conducted on the viscosity, conductivity, and surface tension of the shellac–extract solution using the following instruments: a viscometer (RM 100 CP 2000 Plus, LAMY Rheology, Champagne-au-Mont-d’Or, France), an electrical conductivity meter (Model EC400, Extech Instruments Corporation, Pittsburgh, PA, USA), and a drop shape analyzer (First Ten Angstroms, Portsmouth, VA, USA).

### 2.5. Fabrication of Electrospun Shellac Fibers Loaded with SA Leaf Extract

Each solution was transferred to a 10 mL syringe featuring a needle at the nozzle and connected to a high-voltage power supply during the electrospinning procedure. The fibers that were obtained were deposited on the aluminum foil that was affixed to the rotating cylinder. For all samples, the distance between the needle point and the collector remained constant at 20 cm. The electrospinning procedure was carried out in a controlled environmental setting, characterized by a relative humidity (RH) of 40% to 60% and a temperature range of 23–25 °C.

### 2.6. Entrapment Efficiency of SA Leaf Extract into Electrospun Fibers

The quantity of extract encapsulated within nanofibers was assessed by the quantitative dilution of the rhein content, which was used as an electrospun fiber marker compound. The determination of the entrapment efficiency of the SA leaf extract was performed using the calibration curve of standard rhein. Using serial dilution of standard rhein solutions ranging from 2 g/mL to 30 g/mL, the calibration curve was generated. After dissolving each fiber sample (0.2 g) in ethanol at a concentration of 95% *v*/*v*, the volume was adjusted to 10.0 mL. The rhein concentration was subsequently determined using high-performance liquid chromatography with a diode array detector (HPLC-DAD) on an Agilent 1100 (Agilent Technologies, Santa Clara, CA, USA). The chromatographic separation was performed with an isocratic elution system at a temperature of 40 °C and utilizing a Luna Omega Polar C18 column (100 Å, 5 μm, 4.6 mm × 250 mm) manufactured by Phenomenex Inc., Torrance, CA, USA. A 55:45 (*v*/*v*) ratio was maintained for the mobile phase, which comprised acetonitrile and 0.1% *v*/*v* orthophosphoric acid in an aqueous solution; elution was conducted at a flow rate of 0.6 mL/min. The injection volume, total run duration, and detection wavelength were all configured with the following specific values: 10 µL, 40 min, and 254 nm, respectively. The analysis described above was conducted using the validated method developed by Limmatvapirat C. et al. [11]. The extract’s entrapment efficacy (in %) was determined with the following equation [51].
Entrapment efficiency (%) = (Weight of rhein in the extract entrapped in the fibrous matrix/Weight of rhein in the extract added to the polymer solution) × 100(1)

### 2.7. In Vitro Release Study

The in vitro release study of rhein from SA leaf extract encapsulated in shellac electrospun fibers was investigated using two types of releasing media: phosphate buffer saline (PBS) at pH 7.4 and pH 6.8, with the direct immersion method [52]. To enhance the solubility of the fibers, 1.5% Tween 80 was introduced into each medium. Six fiber samples, including Run 3 (446.89 nm) and Run 18 (426.59 nm), representing large fiber diameters, and Run 19 (277.07 nm) and Run 25 (226.21 nm), representing small fiber diameters, along with the optimized fiber and extract, were utilized to assess and compare the release characteristics of rhein from each sample. Each sample was immersed in 10 mL of dissolution media and shaken at 100 rpm in a temperature-controlled incubator at 37 °C for 12 h. Then, 1 mL of each sample solution was withdrawn at regular time intervals (10 min, 20 min, 30 min, 40 min, 50 min, 1 h, 2 h, 4 h, 6 h, 8 h, 10 h, and 12 h), and an equivalent volume of fresh medium was added to ensure a constant volume. The concentration of rhein released from the collected sample solutions was determined using HPLC-DAD.

### 2.8. Release Kinetics

The dissolution profiles of each fiber underwent fitting with different kinetic models, encompassing the zero-order model, first-order model, Higuchi model, and Korsmeyer–Peppas model, using Excel Add-in DD Solver version 1. The selection of the optimal model relied on criteria such as the lowest Akaike Information Criterion (AIC), the highest Model Selection Criterion (MSC), and the highest adjusted coefficient of determination (R^2^ adj). These criteria collectively guided the identification of the most appropriate model for characterizing the dissolution behavior of the fibers.

### 2.9. Characterization of Electrospun Shellac Fibers Loaded with SA Leaf Extract

#### 2.9.1. Scanning Electron Microscope (SEM)

The morphology of the electrospun shellac fibers containing SA leaf extract was investigated using a scanning electron microscope (SEM) (MIRA 3, Tescan, Brno, Czech Republic). Fiber diameters and the bead-to-fiber ratio were analyzed based on SEM images, using JMicroVision V.1.2.7 software from the University of Geneva, Geneva, Switzerland.

#### 2.9.2. Powder X-ray Diffraction (PXRD)

The investigation of the crystallinity of SA leaf extract, bleached shellac, their physical mixtures, and fibers was conducted utilizing a powder X-ray diffractometer (PXRD) MiniFlex II, manufactured by Rigaku Corporation in Tokyo, Japan. At 40 mV and 30 mA, the analysis was conducted utilizing Cu Kα radiation (λ = 1.5406 Å). To observe whether each sample was crystalline or amorphous, they were scanned at a rate of 4°/min in the 2θ range of 5° to 40°.

#### 2.9.3. Differential Scanning Calorimetry (DSC)

Utilizing a differential scanning calorimeter (DSC 8000, Perkin Elmer, Rodgau, Germany), the thermal properties of electrospun fibers were assessed. Each sample in the aluminum container weighed between 2 and 5 mg. The samples were subsequently heated in the range of 25 °C to 210 °C at a rate of 10 °C/min. An examination was conducted while nitrogen gas was flowing at a rate of 20 mL/min.

#### 2.9.4. Fourier Transform Infrared (FTIR) Spectroscopy 

Using FTIR (Nicolet Avatar 360, Ramsey, MN, USA), the chemical compositions and functional properties of SA leaf extract, bleached shellac, their physical mixtures, and fibers were analyzed. Each sample was incorporated into KBr powder, which was subsequently compacted under hydraulic pressure into a pellet before being inserted into the sample holder. The spectrum was acquired at a resolution of 4 cm^−1^ over the range of wavenumbers from 4000 to 400 cm^−1^.

### 2.10. Antimicrobial Activity of Optimized Fibers

A microbiological technique, the time–kill kinetics assay, is used to assess the antimicrobial effectiveness of a substance within a predetermined time frame. As in a previous study [30], the antibacterial capacity of optimized fibers against *Staphylococcus aureus* ATCC 6538P, *Escherichia coli* DMST 4212, and *Pseudomonas aeruginosa* ATCC 9027 was determined over time. The test microorganisms were subcultured from a suitably isolated colony on an agar plate into a new tube of sterile growth medium, TSB, in order to prepare the bacterial culture. The culture was incubated at 37 °C for 18 to 24 h in a bacterial incubator (Contherm Biosyn 6000CP; Contherm Scientific Ltd., Wellington, New Zealand). Following the incubation period, the microbial culture was evaluated at a wavelength of 600 nm using a UV-vis spectrophotometer (Cary 60, Agilent Technologies, Santa Clara, CA, USA). An absorbance reading between 0.08 and 0.1 was obtained, which corresponds to a bacterial concentration of approximately 0.5 McFarland standard, or 10^8^ CFU/mL. In this test [11], the MIC values for the SA leaf extracts against *S. aureus*, *E. coli*, and *P. aeruginosa* that were determined in the previous broth dilution assay were utilized. The bacterial suspension was diluted to a concentration of 10^6^ CFU/mL. The test samples were sterilized under UV light for 30 min prior to the experiment. Subsequently, they were mixed with bacterial medium to achieve the desired concentration: 3.15 mg/mL for *S. aureus* and 6.25 mg/mL for *E. coli* and *P. aeruginosa*, equivalent to 5 times the minimum inhibitory concentration (MIC) of the SA extract. Extract and bacterial culture without test samples were used as the reference and negative control, respectively. Every sample was incubated at 37 °C in a bacterial incubator. At each designated time interval, 100 µL of the withdrawn sample was diluted to a concentration of 10^5^ CFU/mL before being applied to the sterile agar plate using a spreader. Then, for 18–24 h, all agar plates are incubated at 37 °C to promote colony formation. After the incubation period, the colonies on each plate were counted, and the percentage of viable bacterial cells was calculated using the following equation:The percentage of viable bacterial cells = (C_t_ × 100)/C_0_
(2)

In this context, the microbial loads denoted as C_0_ and C_t_, respectively, are expressed in CFU/mL for the initial time point and each time point specified.

## 3. Results

### 3.1. Extraction and Phytochemical Analysis of SA Leaf Extract

The SA leaf extract, obtained with ultrasound-assisted extraction (UAE) using 95% *v*/*v* ethanol, yielded 8.45 ± 0.35%. The phytochemical analysis confirmed the presence of various compounds in the SA leaf extract, including anthraquinones, flavonoids, terpenoids, saponins, tannins, and steroids. Notably, rhein, an anthraquinone identified in the extract, exhibited a quenching spot with a retention factor (R*_f_*) of 0.47 using a mobile phase of EtOAC:MeOH:H_2_O (100:17:13, *v*/*v*/*v*), aligning with the R*_f_* of the rhein standard. These identified compounds have been associated with antibacterial activity in various studies [6,8,11].

### 3.2. Design of Experiments

#### 3.2.1. Fractional Factorial Design (FFD)

In accordance with an experimental plan generated using Design Expert 8.0.6 software by Stat-Ease Inc. in Minneapolis, MN, USA, nineteen experimental runs, including three center points, were conducted. The design pattern and response results obtained using an FFD are illustrated in Table 3.

Pareto charts are presented to examine the linear effects and interactions among independent variables. These variables are organized in descending order, from the highest frequency to the lowest frequency, moving from left to right. The Pareto bars are color-coded, with orange indicating positive effects and blue representing negative effects on the dependent variables, respectively.

Figure 1a illustrates that the t-value for the applied voltage and shellac content surpassed the standard t-limit, indicating significant positive effects on fiber diameter. The positive influence on fiber diameter was further confirmed by the favorable interrelationship between these two factors. As shown in Figure 1b, positive and negative correlations in the bead-to-fiber ratio were observed for feed rate and shellac content, respectively. The association between these parameters demonstrated a negative relationship with bead formation. Notably, shellac content emerged as the most significant factor impacting the fiber diameter and bead-to-fiber ratio of the electrospun fibers.

In order to estimate the interaction between independent parameters for each response, two-dimensional (2D) contour diagrams are presented. As illustrated in Figure 2a, the coexistence of an elevated applied voltage and a substantial shellac concentration may result in the formation of fibers with sizable diameters. The interactive impact of feed rate and shellac content on the bead-to-fiber ratio is illustrated in Figure 2b. Bead formation was caused by the rapid input rate of the solution and the small quantity of shellac. 

A normal plot of residuals was used to ensure that the chosen model for the experiment adequately fits the data. Figure 3 illustrates that the data values were normally distributed, as evidenced by the straight line. The analysis of variance (ANOVA) in Table 4 reveals that the mathematical models for all response variables were deemed significant, with *p*-values lower than 0.05. Regression coefficient (R^2^) values of 0.8683 and 0.9282 suggest that the resulting models are both predictable and suitable for all dependent variables. The pred R-squared (pred R^2^) values of 0.8274 for fiber diameter and 0.9016 for bead-to-fiber ratio closely aligns with their adjusted R-squared (adj R^2^) values of 0.8420 and 0.9138, respectively. These results affirm the appropriateness of the models for predictions. Furthermore, the absence of significant lack-of-fit values supports the notion of a well-fitted model.

Using multiple regression analysis, the following equations were developed for each response factor:Fiber diameter = 454.82 + 100.39 (X_1_) + 77.99 (X_3_) + 79.03 (X_1_X_3_)(3)
Bead-to-fiber ratio = 1.30 − 1.36 (X_1_) + 0.86 (X_4_) − 0.88 (X_1_X_4_)(4)

#### 3.2.2. Box–Behnken Design (BBD)

Initially, the factors that influence the occurrence of bead formation and fiber diameter were determined using an FFD. Subsequent to this preliminary assessment, the experimental levels of shellac content, SA extract content, applied voltage, and feed rate were chosen for optimization utilizing BBD. Table 5 contains the design matrix and results of multiple responses.

The statistical significance of all response variables was evident from the ANOVA results, which are presented in Table 6. A *p*-value below 0.05 indicated that the mathematical models that were applied to these variables were effective. All regression coefficients had R^2^ values of 0.8406, 0.9780, and 0.8259, indicating that the resulting models were suitable for all dependent variables and were both predictable. The pred R^2^ values, which were in close agreement with their adj R^2^ values (0.7835 for fiber diameter, 0.9546 for bead-to-fiber ratio, and 0.7052 for entrapment efficiency), provided additional substantiation regarding the predictive dependability of these models. The results were substantiated by the lack of substantial lack-of-fit values, which suggested that the models were well-fitted.

Using multiple regression analysis, the following equations were derived to represent all response factors:Fiber diameter = 322.84 + 69.88X_1_ – 40.78 X_3_ + 74.40X_3_^2^(5)
Bead-to-fiber ratio = 0.36 – 1.18X_1_ – 0.12X_2_ – 0.62X_3_ + 0.60X_1_X_3_ + 0.76X_1_^2^ + 0.41X_3_^2^(6)
Entrapment efficiency = 84.25 – 14.08 X_1_ – 4.98 X_2_ + 6.02 X_3_ – 4.96 X_1_^2^ + 5.36 X_3_^2^(7)

The purpose of the normality of the residuals plot is to assess whether the selected model is suitable for the experimental data. The data points exhibit a normal distribution, as demonstrated by the straight-line pattern that is observed (see Figure 4). The observed adherence to a normal distribution provides additional evidence for the validity and sufficiency of the chosen model in characterizing the empirical data.

The impact of distinct variables on each response is illustrated graphically in Figure 5. Shellac content and applied voltage were found to be significant determinants in influencing fiber diameter. The correlation between increased shellac content and the production of larger fiber diameters is illustrated in Figure 5a. The obvious effect of varying voltage levels on fiber diameter is demonstrated in Figure 5b. At the onset, the application of low voltage induced a dilation in the fiber. A subsequent observation of fiber size reduction can be made at a voltage level of medium magnitude. Nonetheless, the implementation of elevated voltage resulted in a recurrence of fiber diameter enlargement. With both lower and higher levels of applied voltage and an increase in shellac content, the diameter of the filaments exhibits a discernible enlargement. These observations underscore the susceptibility of fiber diameter to changes in the amount of shellac utilized and the voltage applied, providing valuable insights into the intricate relationship between these variables and the resulting fiber characteristics.

The distinct effects of individual parameters on the bead-to-fiber ratio are shown in detail in Figure 5c–e. There was a correlation observed between increased levels of shellac content, extract content, and applied voltage and a decrease in the bead-to-fiber ratio, which indicates a reduction in the formation of beads. On the contrary, reduced voltage and diminished concentrations of shellac and extract were recognized as variables that promote the formation of beads.

In terms of entrapment efficiency, elevated values were noted under particular conditions, such as reduced amounts of shellac and extract, coupled with a heightened applied voltage, as depicted in Figure 5f–h.

In order to forecast the relationship between responses and independent parameters, 2D contour diagrams were produced. The correlation between the quantity of shellac and the voltage had a substantial impact on the ratio of beads to fibers. The results illustrated in Figure 5i demonstrate that the formation of beads was facilitated by a low voltage and a small quantity of shellac. Conversely, for medium to high voltage and high shellac content, the bead-to-fiber ratio was minimized. The aforementioned results underscore the intricate interconnections among the independent variables and their influence on the intended outcome.

The optimization criteria for electrospinning conditions, with an emphasis on desirable ranges for all response variables, are succinctly outlined in Table 7 for the purpose of verifying the optimized conditions. The feed rate was consistently maintained at 0.8 mL/h due to the lack of statistical significance observed in any of the responses. In order to identify the region where the predicted values of all response variables remained within permissible ranges, an overlay contour plot (Figure 6) was produced. The design space, denoted by the yellow region on the graph, comprises optimal conditions that satisfy the desirability criteria for multiple responses. Table 8 delineates the optimal conditions that govern the desirable ranges of the three responses. In order to authenticate these conditions, further experiments were performed in triplicate. The verification results, which are comprehensively presented in Table 9, provide additional evidence that the observed values fell within permissible ranges, thereby bolstering the dependability of the optimized electrospinning conditions.

Under optimized electrospinning conditions, the resultant fibers demonstrated a nanosized morphology devoid of beads, featuring a diameter of 306 nm and a low bead-to-fiber ratio of 0.29. In comparison with earlier studies, where electrospinning of shellac and monolaurin resulted in larger fibers with a diameter of 488 nm and a bead-to-fiber ratio of 0.48 under specific conditions [23], and another investigation incorporating shellac loaded with *Kaempferia parviflora* extract also exhibited larger fiber diameter and bead formation [30], our ultrafine fibers exhibited a unique profile characterized by a smaller diameter and a notable reduction in bead formation. This divergence in morphology can be attributed to the utilization of bleached shellac in our study, distinguished by its superior purity obtained with a bleaching process. The inherent lower viscosity of the electrospinning fluid, a result of the use of bleached shellac, contributed to the production of smaller diameter fibers with a tendency toward bead formation. However, it is important to note that the inclusion of SA leaf extract played a pivotal role in modulating the solution’s viscosity and conductivity, effectively counteracting the low viscosity associated with bleached shellac. This adjustment proved instrumental in minimizing bead formation along the fibers.

In summary, the distinctive characteristics of the electrospun fibers in this study, namely, their smaller diameter and reduced bead formation, can be attributed to the strategic use of bleached shellac and the incorporation of SA leaf extract. These modifications not only prevented clogging issues related to lower viscosity but also facilitated the attainment of the desired ultrafine fiber morphology.

### 3.3. Evaluation of Solution Properties

The morphology of electrospun fibers is significantly influenced by the characteristics of the electrospinning fluids. This study aimed to assess the impact of viscosity, surface tension, and conductivity in relation to various shellac–extract ratios. Table 10 provides a comprehensive overview of the measurements conducted, revealing noteworthy trends. According to the results, viscosity displayed an upward trend with increasing concentrations of both shellac and extract. Specifically, higher levels of shellac and extract content were correlated with elevated viscosity values. Concurrently, the conductivity of the shellac solution generally decreased with higher concentrations of shellac. Interestingly, the addition of extract at the same shellac concentration resulted in a marked increase in electrical conductivity as the extract content rose. In contrast, surface tension values exhibited minimal variation across different ratios of shellac and extract. The consistent nature of surface tension suggests that changes in shellac and extract content have a limited effect on this particular parameter. In summary, this study underscores the significant influence of electrospinning fluid characteristics on fiber morphology, with viscosity, conductivity, and surface tension serving as key factors in shaping the electrospun fibers. The findings provide valuable insights into the intricate relationship between shellac–extract ratios and the resulting properties of electrospun fibers.

### 3.4. Entrapment Efficiency of Electrospun Shellac Fibers Loaded with SA Leaf Extract

The quantity of entrapped rhein in the SA leaf extract within nanofibers was ascertained using a validated HPLC-DAD method for quantitative analysis of rhein content [11]. The determination of the amount of rhein contained in the fibers was performed using a linear regression equation and a coefficient of determination (R^2^ = 0.9992) derived from the calibration curve of standard rhein. The retention time of the rhein peak at 23.5 min is identical in the HPLC chromatograms of shellac fibers laden with SA extract (a) and standard rhein (b), as shown in Figure 7. As detailed in Table 5, the entrapment efficiency, as assessed under diverse electrospinning conditions, varies between 58.88% and 105.42%. These results highlight the considerable impact of diverse electrospinning conditions on the entrapment efficiency of SA leaf extract within shellac fibers. The ability to discern the rhein peak in the chromatograms further reinforces the accuracy of the HPLC-DAD method in quantifying the entrapped rhein, providing valuable insights into the efficiency of the electrospinning process in encapsulating the active compound within the nanofibers.

### 3.5. In Vitro Release Study

In the pH 7.4 releasing medium, the release patterns of all fiber samples generally align, as depicted in Figure 8a. An initial release of rhein ranging from 63% to 88% was observed within the first hour across all fiber samples. The maximum cumulative release among all the samples was recorded as follows: 90.95% for Run 3 (large-diameter fibers), 91.72% for Run 18 (large-diameter fibers), 88.94% for Run 19 (small-diameter fibers), 92.88% for Run 25 (small-diameter fibers), 94.50% for the optimized fibers, and 81.61% for the intact extract. The results suggested an enhancement in release after incorporating the extract into the shellac fibers.

Additionally, the release behaviors were characterized by two distinct phases: an initial burst release phase, followed by a sustained release phase. After the initial burst release, the percentage of rhein released from each fiber sample reached a maximum and remained constant from 4 h to 12 h. During this period, the remaining rhein molecules were released at a relatively constant rate over an extended period, especially those from larger fiber diameters.

As shown in Figure 8b, rhein could not be completely released from all fibers in the medium at a pH of 6.8. Consequently, the cumulative release percentage of rhein from all fiber samples in pH 7.4 was higher than in the pH 6.8 media. After 24 h of immersion, all fibers and the extract were completely dissolved at pH 7.4, whereas not all fiber samples exhibited complete solubility at pH 6.8. The presence of undissolved fibers in media at pH 6.8 can be attributed to this pH being lower than the dissolution pH of shellac. The cumulative release of the intact extract did not show significant variations between both pH conditions. 

The findings underscore the pH-dependent release behavior of rhein from electrospun fibers, emphasizing the importance of considering environmental conditions in designing drug delivery systems. This information contributes valuable insights to the field of controlled drug release and formulation design.

### 3.6. Release Kinetics

In order to investigate the mechanism of rhein release from fibers, a number of kinetic models (including zero-order, first-order, Higuchi, and Korsmeyer–Peppas) were applied to the release data obtained from the pH 7.4 and 6.8 media. Based on the lowest AIC, highest MSC, and highest adj R^2^ values, the most suitable fitting model was identified. In particular, Run 19 and Run 25 (small-diameter fibers) and optimized fibers exhibited the most accurate zero-order release kinetics in the pH 7.4 medium, as indicated by the R^2^ values of 0.9217, 0.9722, and 0.9604, respectively. In contrast, the first-order release kinetics of Run 3 and Run 18 (large-diameter fibers) were characterized by R^2^ values of 0.9741 and 0.9558, respectively. The sustained release pattern that minimizes explosive release and attains a constant drug concentration within the therapeutic range is the zero-order release pattern [53].

After fitting the release data acquired from PBS with a pH of 6.8 to various mathematical models, it was determined that the first-order kinetics model provided the most accurate representation for small-diameter fibers and optimized fibers. This implies that the quantity of drug discharged is directly proportional to the quantity of drug remaining in the matrix [54]. Large-diameter fibers, on the other hand, demonstrated the Korsmeyer–Peppas model. Run 3 exhibited a release exponent (n) of 1.279, which suggests the presence of a Super Case II transport mechanism. With a release exponent (n) of 0.730, anomalous or non-Fickian behavior was observed in Run 18 (large-diameter fibers). This observation implies that drug release is influenced by a combination of diffusion and attrition mechanisms, which is suggestive of a multifaceted drug release pattern [55].

This comprehensive analysis of release kinetics provides valuable insights into the diverse behaviors exhibited by different fiber types in response to varying pH conditions. Understanding these mechanisms is crucial for tailoring drug delivery systems to achieve desired release profiles for enhanced therapeutic efficacy.

### 3.7. Characterization of Electrospun Shellac Fibers Loaded with SA Leaf Extract

#### 3.7.1. Scanning Electron Microscope (SEM)

SEM was used to scrutinize the morphology of electrospun shellac fibers loaded with SA leaf extract. Table 5 presents the fiber diameter and bead-to-fiber ratio, as determined using the SEM images. These results were visually reinforced by the SEM images in Figure 9 and Figure 10. Figure 9a,b showcases SEM images of fibers at varying concentrations of shellac under consistent electrospinning conditions, revealing a noticeable increase in fiber diameters with higher shellac content. Figure 9c,d illustrates that under identical shellac: extract ratios and flow rates, significant variations in fiber diameter were observed across different applied voltages. Notably, an increase in applied voltage led to a reduction in fiber diameter compared with lower voltage settings. This reduction in fiber diameter with a higher applied voltage suggests that electrospinning under elevated voltage settings could be a promising strategy for achieving finer and more controlled fibers. The relationship between applied voltage and fiber diameter merits further investigation, as it may have implications for the optimization of electrospinning processes for the fabrication of shellac fibers loaded with SA leaf extract.

In Figure 10a,b, the SEM images vividly illustrate the influence of shellac concentrations (35% *w*/*w* and 45% *w*/*w*) on the bead-to-fiber ratios under identical electrospinning conditions. It is evident that lower shellac content enhances the formation of beads along the fibers. Figure 10c,d further highlights the impact of varying extract content on bead formation. Here, it is observed that a lower extract content results in fibers with a higher occurrence of beads, showcasing the sensitivity of the electrospinning process to extract concentration. Exploring the effect of voltage on the bead-to-fiber ratios, Figure 10e,f reveals a substantial difference between high and low-voltage applications. Notably, lower voltage settings generate a greater number of beads compared with higher voltage, indicating that the electrospinning process’s electric field strength significantly influences bead formation.

These findings underscore the importance of meticulous control over processing parameters, such as shellac and extract concentrations, as well as voltage settings, in achieving desired fiber morphology. Understanding these relationships is crucial for tailoring electrospinning conditions to produce fibers with specific characteristics, thereby advancing the development of functional materials in various applications.

#### 3.7.2. Powder X-ray Diffraction (PXRD)

Figure 11 presents the PXRD patterns of bleached shellac, SA leaf extract, their physical mixtures, and the optimized electrospun shellac fibers loaded with SA extract. The diffractograms of bleached shellac revealed broad peaks at 2Ɵ° angles of 9.90 ± 0.3°, 18.21 ± 0.14°, and 39.40 ± 0.3°, with corresponding intensities of 218 ± 19, 1288 ± 43, and 358 ± 30, respectively. In the physical mixtures and electrospun fibers, these broad peaks were observed around 2Ɵ° angles of 18.45 ± 12° and 18.54 ± 12°, characterized by intensities of 1186 ± 34 and 1209 ± 34, respectively. Notably, the PXRD pattern of the SA leaf extract did not provide specific information, suggesting its amorphous nature.

Considering that shellac is a semicrystalline polymer with low crystallinity [56], the observed broad peaks in its PXRD pattern are indicative of an amorphous structure [18,57]. This amorphous nature was also evident in the diffractogram of SA leaf extract, aligning with the overall trend. Importantly, no discernible differences were observed between the physical mixtures and the electrospun shellac fibers loaded with SA extract.

The consistent amorphous nature in both the physical mixtures and electrospun fibers indicates the successful incorporation of SA leaf extract into the shellac matrix. This is essential for potential applications, as amorphous structures are often associated with enhanced solubility and bioavailability of active pharmaceutical ingredients. The findings from the PXRD analysis reinforce the suitability of the electrospinning process for producing composite fibers with uniform and amorphous characteristics, showcasing its potential in controlled drug delivery systems.

#### 3.7.3. Differential Scanning Calorimetry (DSC)

The application of DSC aimed to explore the thermal properties of the fibers, and the outcomes are depicted in Figure 12. The endothermic peak of bleached shellac was observed at approximately 54 °C, aligning with findings from previous studies [18]. In both physical mixtures and fibers, similar peaks were identified at temperatures of 55.75 °C and 55.04 °C, respectively. Interestingly, no distinct peak was discerned in the DSC thermogram of the SA leaf extract. While a broad exothermic peak for the extract appeared at 185.84 °C, this feature was notably absent in the thermogram of the fibers. The presence of this peak could be attributed to either the decomposition of the extract or the complete integration of the SA extract into the fibers. The absence of a distinct peak in the fiber thermogram around the temperature associated with the exothermic peak of the SA leaf extract raises intriguing possibilities. It suggests that the electrospinning process might have led to the effective incorporation of the SA extract into the fibers, potentially altering the thermal characteristics of the extract.

This observation highlights the transformative nature of the electrospinning technique, potentially influencing the thermal behavior of the composite fibers. Further investigations into the structural changes and interactions occurring during the electrospinning process could provide valuable insights into the successful integration of SA leaf extract into the shellac matrix. The DSC results underscore the potential of electrospinning as a method for tailored drug delivery systems, where thermal characteristics play a crucial role in the stability and performance of the final product.

#### 3.7.4. Fourier Transform Infrared (FTIR) Spectroscopy 

The chemical compositions and interactions between shellac and the SA leaf extract subsequent to the production of electrospun fibers were investigated with FTIR. The confirmation of the carboxylic acid group’s presence in both the shellac and extract was achieved by observing the peaks at 1709 cm^−1^ and 1706 cm^−1^, which correspond to the vibration of C=O stretching; 1463 cm^−1^, 1374 cm^−1^, 1247 cm^−1^; and 1448 cm^−1^, 1361 cm^−1^, 1259 cm^−1^, which correspond to C-O stretching, as depicted in Figure 13. At 3400 cm^−1^ and 3301 cm^−1^, respectively, the shellac and extract exhibited prominent peaks that were ascribed to the O-H stretching vibration, which signifies the presence of OH functional groups. In both the shellac and extract, C-H stretching absorption took place at 2928 cm^−1^ and 2856 cm^−1^, and 2919 cm^−1^ and 2850 cm^−1^, respectively. The identification of distinct peaks in the bleached shellac and extract was in accordance with findings reported in the literature [18,57,58,59,60,61]. 

As depicted in Figure 13, the FTIR spectrum of the rhein-rich extract exhibits broad bands corresponding to hydroxyl groups (at 3301 cm^−1^), medium peaks associated with C-H stretching (at 2919 cm^−1^ and 2850 cm^−1^), paired C=C-C stretching bands within aromatic rings (at 1600 cm^−1^ and 1448 cm^−1^), strong peaks indicative of conjugated carbonyl groups (at 1706 cm^−1^), and medium peaks related to C-O groups (at 1039 cm^−1^). These specific peaks in the spectrum of the SA leaf extract align with the structural features of rhein (Figure 14) and corroborate previous findings [11]. This consistency strongly suggests that rhein constitutes the primary component of the extract.

Due to the significant quantity of shellac present in the shellac–extract ratio, the physical mixtures exhibited a nearly identical FTIR spectrum to that of bleached shellac. However, following the development of electrospun fibers loaded with SA extract, the majority of the extract’s absorption peaks shifted to the higher frequency region: 3301 cm^−1^ to 3381 cm^−1^, 2919 cm^−1^ to 2928 cm^−1^, 2850 cm^−1^ to 2856 cm^−1^, 1706 cm^−1^ to 1710 cm^−1^, 1600 cm^−1^ to 1635 cm^−1^, and 1448 cm^−1^ to 1463 cm^−1^. The fiber spectrum undergoes a shift as a result of hydrogen bonding between the rhein compound present in the extract and the shellac.

This shift in the fiber spectrum reflects the successful integration of SA leaf extract into the shellac matrix during the electrospinning process, with potential implications for the functional properties of the resulting fibers. The findings underscore the feasibility of tailoring fiber compositions for specific applications using controlled electrospinning processes, providing a foundation for further exploration in drug delivery and related fields.

### 3.8. Antimicrobial Activity of Optimized Fibers

The time–kill study is a widely recognized microbiological evaluation method that provides essential insights into the time-dependent pharmacodynamics of antimicrobial substances, assessing their bactericidal activity. In this study, the antibacterial kinetics of the optimized fibers were compared to those of the intact extract. The time–kill profiles for three distinct bacterial species—*S. aureus*, *P. aeruginosa*, and *E. coli*—are illustrated in Figure 15. Each data point signifies the percentage of viable bacterial organisms at specific time intervals (*n* = 3). Notably, a significant reduction in viable bacterial populations was observed in both the optimized fibers and the SA extract after a 9-h incubation period for all microorganisms. The antimicrobial activities of optimized fibers are shown in Figure 16.

Within a 9-h incubation period, the optimized fibers loaded with SA extract demonstrated a substantial reduction in viable bacterial cells for *S. aureus*, reaching 23.01% compared with the bacterial control (Figure 15a). In particular, the optimized fibers had better antimicrobial activity, as shown by lower percentages of viable bacterial cells for both *P. aeruginosa* (17.70%) and *E. coli* (13.88%) compared with the SA extract after 9 h of incubation (Figure 15b,c). The reduced proportions of viable bacterial cells observed in the optimized fibers indicate enhanced antimicrobial effectiveness. This study’s findings highlight that the optimized fibers loaded with SA extract displayed superior inhibitory effects against *P. aeruginosa* and *E. coli* after a 9-h incubation period compared with the SA extract alone. An initial surge release of rhein was noted within the first hour of fiber loading with the SA extract, as depicted in Figure 8. This observation aligns with the results from the optimized fibers, which exhibited a considerably greater reduction in viable cell count following a 1-h incubation period against each pathogen. Furthermore, as the incubation period extended to 9 h, there was a gradual reduction in the percentage of viable bacterial cells (Figure 15 and Figure 16). All of these results show that optimized fibers loaded with SA extract could be used as an effective wound dressing material, showing longer-lasting and stronger antimicrobial activity against a wide range of bacterial strains. 

## 4. Discussion

The outcome of electrospinning a polymer solution, including the structure and morphology of the nanofibers, is typically influenced by polymer-specific solution parameters, processing conditions, and ambient factors. Critical factors that influence the spinnability of a polymer solution include its concentration and electrical conductivity. The examination of the impact of applied voltage and shellac content on fiber diameter has yielded valuable insights into the electrospinning procedure. A correlation was observed between greater shellac concentration and the formation of larger fiber diameters; this was attributed to the increased viscosity and solution entanglement caused by the higher shellac concentration. An increase in the shellac content hinders the elongation resistance of the electrospinning process, leading to the generation of fibers with larger diameters [62].

The adjustment of voltage levels significantly influenced the diameters of the fibers. At first, the implementation of low voltage resulted in an expansion of fiber diameter as a consequence of diminished electrostatic forces. This caused the polymer flow to experience reduced stretching, ultimately producing fibers with larger diameters. Following that, a decrease in fiber diameter was detected at a medium voltage level, possibly as a result of balanced electrostatic forces, which caused the polymer jet to stretch optimally and produce fibers with a smaller diameter. On the contrary, the implementation of elevated voltage resulted in an enlargement of the fiber diameter. This finding suggests that an excess of electrostatic forces may cause the polymer flow to undergo overstretching, thereby producing fibers with larger diameters [62,63]. Although certain studies have reported no correlation between the voltage applied and the diameter of the fibers [64], it was discovered that increasing the voltage during the electrospinning process generated more charges, which improved the polymer solution’s ability to stretch and contributed to the production of thinner nanofibers with a smaller diameter [65]. These results are consistent with those of other investigations [23,66,67] that have documented comparable results.

The observed correlation between varying levels of shellac content, extract content, and applied voltage with the bead-to-fiber ratio sheds light on the intricate dynamics of the electrospinning process. High levels of shellac content, extract content, and applied voltage were associated with a reduction in the bead-to-fiber ratio, indicating a tendency for smoother fiber formation under these conditions. The presence of fewer beads in conditions of higher shellac content can be attributed to the increased viscosity and improved solution entanglement. This impedes bead formation, favoring the development of smoother fibers. The elevated shellac content likely alters the balance between viscosity and surface tension, promoting a smoother electrospinning process. In contrast, lower shellac and extract content, as well as low voltage, were linked to enhanced bead formation. The lower shellac concentration results in reduced viscosity, allowing surface tension to exert a more dominant influence along the electrospinning jets. This dominance of surface tension, coupled with fewer chain entanglements, contributes to bead formation along the fibers [62]. These findings suggest that the interplay between shellac concentration, extract content, and applied voltage significantly influences the morphology of electrospun fibers. Understanding these relationships provides valuable insights for optimizing the electrospinning process to achieve specific fiber characteristics, enhancing the potential applications of electrospun materials.

An elevated concentration of extract may potentially enhance the conductivity of the solution, thereby promoting the expansion of the polymer stream and reducing the formation of beads. A substantial increase in solution conductivity was observed subsequent to the introduction of the extract. In this context, the carboxylic acid composition of the extract is noteworthy. Compounds possessing carboxylic groups, rhein, in particular, are capable of ionization, which produces charged species (ions) that enhance the conductivity of the solution [68]. Furthermore, the inclusion of substantial quantities of the extract resulted in an increase in the viscosity of the electrospinning solutions. This phenomenon may potentially explain the decreased amount of bead formation. On the contrary, reduced concentrations of shellac and extract, along with low voltage, can cause a decline in the viscosity and conductivity of the solution. This, in turn, can contribute to inadequate polymer jet elongation and an increase in the formation of beads. Moreover, elevating the applied voltage can enhance the electrostatic forces exerted on the polymer flow, which facilitates the development of a more uniform and elongated fiber structure, consequently diminishing the occurrence of beads [69]. These findings underscore the critical influence of extract concentration and processing parameters on the electrospinning process, providing valuable insights for optimizing conditions to achieve desired fiber morphologies. Effective control of these variables can lead to improved electrospinning outcomes for various applications in materials science.

A high degree of entrapment efficiency is discernible under particular circumstances, which involves the application of a high voltage and a low concentration of shellac and extract. These results indicate that the aforementioned conditions might augment the entrapment of the intended components within the fibers. It is probable that decreased concentrations of shellac and extract lead to enhanced solubility and decreased solution viscosity, thereby facilitating the successful incorporation of the extract into the polymer matrix. Conversely, as the concentration of both shellac and extract increased, a significant reduction in the efficiency of entrapment was detected. The potential cause of this decrease could be the existence of immobile constituents in the solution or the substances’ poor solubility [70]. These findings provide valuable insights into the interplay between solution composition and entrapment efficiency, offering guidance for optimizing electrospinning conditions to achieve effective incorporation of components within the fibers for various applications in materials science.

The implementation of elevated voltage emerges as a potential strategy to enhance the efficacy of entrapment in the electrospinning process. This observed phenomenon can be attributed to the intensified electrostatic forces and elongation of the polymer jet during electrospinning. The heightened capacity for elongation likely results in improved entrapment and the incorporation of the target substances within the fibers. Moreover, the production of smoother, more uniform fibers, a consequence of elevated applied voltage, contributes to enhanced entrapment efficiency. The significant surface area of the polymers generated under high voltage facilitates the passive loading of the substance, further contributing to the increased level of entrapment [71,72]. In conclusion, the application of elevated voltage in the electrospinning process holds promise for enhancing the efficacy of entrapment. This improvement is attributed to intensified electrostatic forces, increased polymer jet elongation, and the production of smoother fibers, collectively leading to improved entrapment efficiency. The substantial surface area of the polymers created under elevated voltage also facilitates passive loading of the substance, solidifying the entrapment of the extract within the matrix. These findings underscore the importance of carefully managing processing parameters, specifically voltage, to optimize entrapment outcomes in electrospinning applications, offering valuable insights for various material science applications.

The optimized electrospun shellac fibers laden with SA leaf extract exhibited two distinct phases of release behavior in releasing media with pH values of 7.4 and 6.8: an initial burst release phase followed by a sustained release phase. A burst release may transpire when a fraction of the substance or compound is released rapidly from the surface or in close proximity to the surface region of the fibers. The burst release mechanism is commonly linked to molecules that are either loosely affixed to the fiber matrix or exist on its surface. This process facilitates an instantaneous discharge of the compound. After the initial explosive release, the rate of rhein release from each fiber sample peaked at 4 h and remained constant for the next 12 h, indicating a sustained release. Throughout this prolonged period, the residual rhein molecules were discharged at a relatively consistent rate. Typically, the sustained release was attributed to the drug or compound diffusing across the fiber matrix. A constant release rate is maintained as the molecules are continuously replenished from within the matrix as they diffuse out of the fibers [73]. Additionally, the relatively slow dissolution rate of shellac, influenced by water absorption and expansion, further contributes to the formation of sustained release profiles, emphasizing the suitability of these fibers for controlled drug release applications [53].

The optimized fibers demonstrated the highest percentage of rhein release, surpassing the release rates of the other fiber samples. Notably, the optimized fibers exhibited a faster release of rhein compared with both smaller- and larger-diameter fibers, achieving maximum release within 4 h. In contrast, smaller-diameter fibers peaked in release within 6 h, while larger-diameter fibers showed the highest percentage of release after 8 h. The observed results can be attributed to the impact of fiber diameter on drug release kinetics. Smaller-diameter fibers exhibited a faster release, reaching a peak release within a shorter timeframe of 6 h. This phenomenon is linked to the increased surface area-to-volume ratio, which enhances the contact area between drug-loaded fibers and the surrounding medium. The elevated surface area promotes a higher rate of drug release [74]. On the contrary, larger-diameter fibers, with reduced porosity and a lower surface area-to-volume ratio, demonstrated a slower release profile. The prolonged diffusion path and limited contact with the surrounding medium contributed to a delayed release of the drug. The optimization process applied to create the optimized fibers was specifically aimed at maximizing surface area and diffusion characteristics. This optimization likely accounts for the comparatively faster release observed in these fibers compared with others. These findings emphasize the critical role of fiber diameter in governing drug release kinetics in electrospun fibers, providing valuable insights for tailored drug delivery applications.

In the context of pH-dependent release, the cumulative release of rhein from all fiber samples at pH 6.8 was notably diminished compared with pH 7.4. This discrepancy can be attributed to the comparatively higher dissolution pH of shellac (around 7.3) and its poor solubility in aqueous solutions, resulting in limited rhein release at pH 6.8 [75]. Conversely, at pH 7.4, where shellac exhibits full solubility, a higher cumulative release was observed [58,76]. The extract demonstrated complete solubility under both pH conditions. However, the release rate of rhein from the extract was substantially slower compared with the fibers loaded with the extract. This discrepancy suggests that fiber-mediated drug release could occur at an accelerated rate, potentially influenced by polymer degradation, hydration, or drug diffusion processes [77]. The amorphous state of the extract and nanometer-sized diameters of the fibers significantly increased the rate of rhein release in PBS solution (pH 7.4) in comparison with the extract in its intact state. In addition, it should be noted that biphasic release profiles, which comprise an initial rapid release followed by a sustained release, may prolong the efficacy of bacterial inhibition [78]. This unique release pattern further underscores the versatility and potential therapeutic benefits of the electrospun shellac fibers loaded with SA leaf extract.

The characterization studies provide strong evidence of the successful incorporation of the SA leaf extract into the electrospun shellac fibers. The FTIR spectrum of the optimized fibers loaded with SA leaf extract revealed a notable shift in the majority of the extract’s absorption peaks to higher frequency regions. This shift, particularly in OH stretching, C-H stretching, C=O stretching, and C=C-C stretching bands within aromatic rings, suggests a significant interaction between rhein (the primary component of the extract) and shellac within the fibers. This interaction is crucial as it signifies the compatibility and integration of the extract into the polymeric matrix, potentially influencing the release kinetics and overall performance of the fibers.

The PXRD patterns of both the shellac and extract, characterized by the absence of diffraction peaks, indicate their amorphous states. Interestingly, this amorphous characteristic was maintained in the electrospun shellac fibers loaded with SA leaf extract. This consistency in amorphous structure suggests that the electrospinning process did not adversely impact the amorphous nature of shellac and extract. The amorphous state is often desirable in drug delivery systems, as it can enhance solubility and dissolution rates.

Given that the SA leaf extract demonstrated antimicrobial effectiveness against certain pathogens, the successful integration of the extract into the electrospun fibers opens up avenues for assessing the potency of these extract-loaded electrospun shellac fibers. Further evaluation will be crucial to determining the antimicrobial efficacy of the fibers and their potential applications in targeted drug delivery and infection control. A significant reduction in viable microbial cells was consistently observed at every incubation time point throughout the study on time–kill kinetics when both the optimized fibers and the SA extract were utilized. Distinct differences were observed in the viable bacterial reduction rates between the fibers and the extract, as determined by the analysis. Remarkably, it became evident within the first hour that the SA leaf extract demonstrated a more expeditious decline in the microbial population in comparison with the optimized fibers. The observed discrepancies can potentially be attributed to differences in the release characteristics of the SA leaf extract and the optimized fibers. Prior studies have indicated a potential correlation between the antibacterial activities of electrospun fibers and the drug-release behaviors exhibited by them [74]. A potential reason for the observed rapid decrease in bacterial viability within the initial hour is that the extract may have a more immediate and direct effect. On the contrary, the optimized fibers have the potential to facilitate an extended and sustained antibacterial effect, which could lead to a more significant decrease throughout the nine-hour duration. The results of this study indicate that the optimized fibers successfully control the incremental discharge of antimicrobial agents from the matrix during the incubation phase. The implementation of controlled release may potentially extend the bacterial population’s exposure to antimicrobial compounds, specifically rhein, which could lead to a gradual yet consistent decline in the viable bacterial population. Furthermore, the observed variations in antimicrobial effectiveness against distinct bacterial strains may be attributed to various factors, including the composition and concentration of active compounds in the SA leaf extract, as well as particular interactions between the fiber matrix and bacterial cells. While the antimicrobial effect between the extract and the optimized fibers did not exhibit a significant difference, this can be attributed to the rapid initial dissolution of the extract. However, toward the end of the incubation time, the optimized fibers demonstrated a higher release of rhein, contributing to a more effective reduction in microbial loads. Despite this, the form of the optimized fibers proved to be more convenient, facilitating better air passage and promoting optimal wound healing. This convenience stands as a distinct advantage of the fiber patch. Additional research is required to elucidate the fundamental mechanisms of these materials in order to yield practical implications for antimicrobial therapies. Furthermore, it is necessary to investigate the wider use of these fibers in the field of wound care in order to maximize their effectiveness in clinical environments.

## 5. Conclusions

The morphology of electrospun nanofibers is primarily influenced by shellac concentration, followed by applied voltage and extract content. These observations highlight the susceptibility of the fiber diameter to variations in shellac quantity and applied voltage, providing insights into the complex interplay among these factors and resultant fiber properties. Electrospinning conditions with high values of applied voltage, shellac, and extract contents effectively reduce the bead-to-fiber ratio. Shellac content, extract content, and voltage were identified as influencing factors on entrapment efficiency, with favorable results achieved using the conditions of high voltage and low contents of shellac and extract. Optimal conditions (38.5% *w*/*w* shellac content, 3.8% *w*/*w* extract content, and an applied voltage of 24 kV) were established to achieve a nanosized fiber diameter (306 nm), a low bead-to-fiber ratio (0.29), and a 96% entrapment efficiency of extract into fibers. The in vitro release study of the optimized nanofibers revealed a biphasic pattern, with an initial burst release accounting for 88% within the first hour, followed by sustained release surpassing 90% up to 12 h. Furthermore, this study unveiled the antimicrobial efficacy of optimized fibers infused with SA extract against *S. aureus*, *P. aeruginosa*, and *E. coli*. Notably, these fibers exhibited precise control over the gradual release of rhein, maintaining a sustained reduction in viable bacterial count over the entire 9-h incubation period. These findings underscore the potential synergies achievable by combining electrospun shellac fibers with SA extract for antimicrobial applications. Further investigation into the mechanisms and optimization of these materials holds promise for innovative and effective strategies in the ongoing battle against bacterial infections.

## Figures and Tables

**Figure 1 polymers-16-00183-f001:**
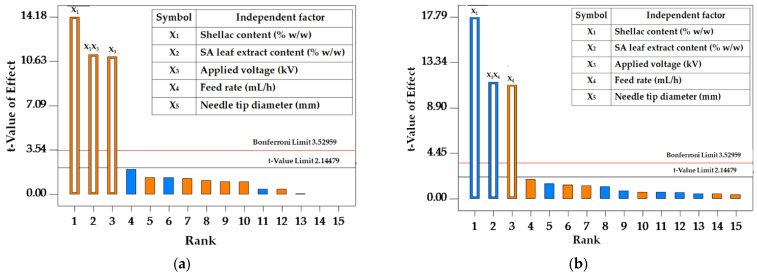
Pareto chart showing the effect of parameters on fiber diameter (**a**) and the bead-to-fiber ratio (**b**). The orange bars depict positive relationships, while the blue bars illustrate negative relationships.

**Figure 2 polymers-16-00183-f002:**
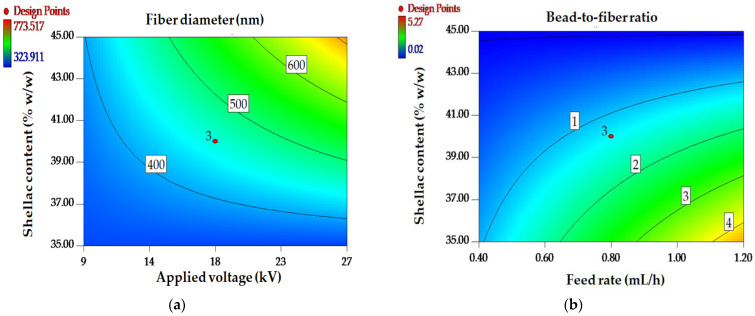
Contour plots showing the interactions between factors on fiber diameter (**a**) and the bead-to-fiber ratio (**b**).

**Figure 3 polymers-16-00183-f003:**
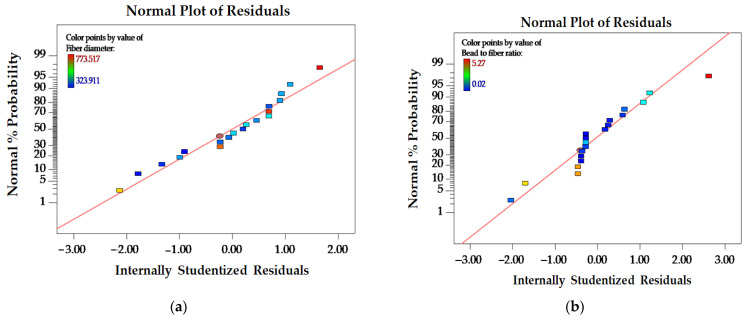
Plot of the internally studentized residual’s normal probability for fiber diameter (**a**) and the bead-to-fiber ratio (**b**).

**Figure 4 polymers-16-00183-f004:**
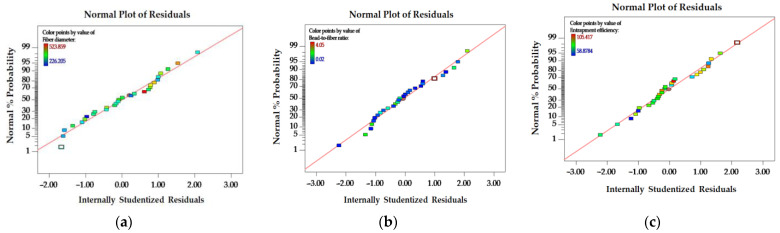
Normal probability plot of the internally studentized residuals for fiber diameter (**a**), the bead-to-fiber ratio (**b**), and entrapment efficiency (**c**).

**Figure 5 polymers-16-00183-f005:**
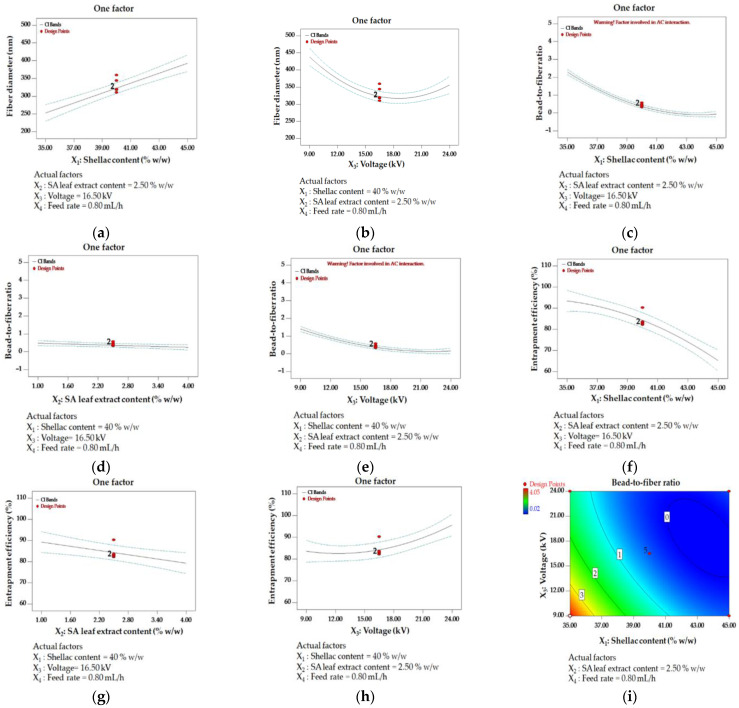
One factor affecting multiple responses: fiber diameter influencing shellac content (**a**) and applied voltage (**b**); the bead-to-fiber ratio influencing shellac content (**c**), extract content (**d**), and applied voltage (**e**); entrapment efficiency influencing shellac content (**f**), extract content (**g**), and applied voltage (**h**); and a contour plot illustrating the interaction between shellac content and applied voltage on the bead-to-fiber ratio (**i**).

**Figure 6 polymers-16-00183-f006:**
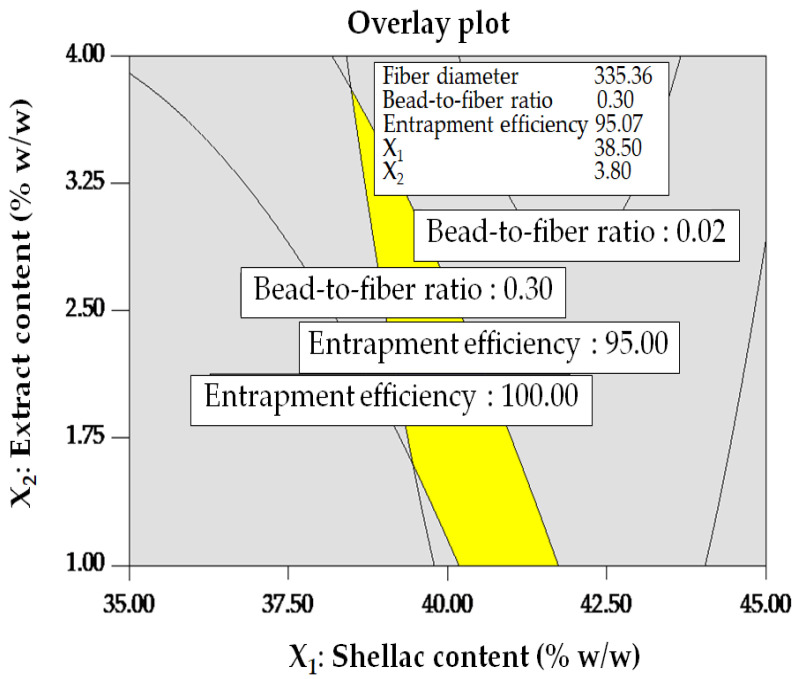
Overlay contour plot for optimized conditions. The region that meets the criteria is highlighted in yellow, while the area that does not meet the criteria is displayed in gray.

**Figure 7 polymers-16-00183-f007:**
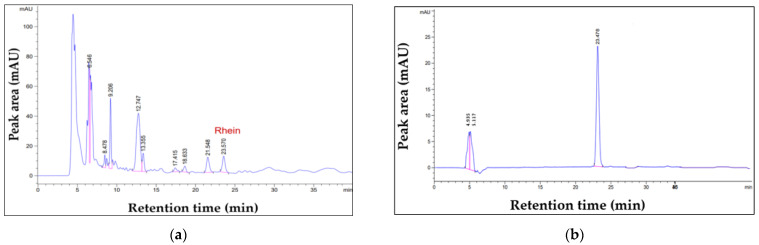
The HPLC chromatograms of shellac fibers loaded with SA extract (**a**) and the standard rhein (**b**) reveal an identical retention time for the rhein peak at 23.5 min.

**Figure 8 polymers-16-00183-f008:**
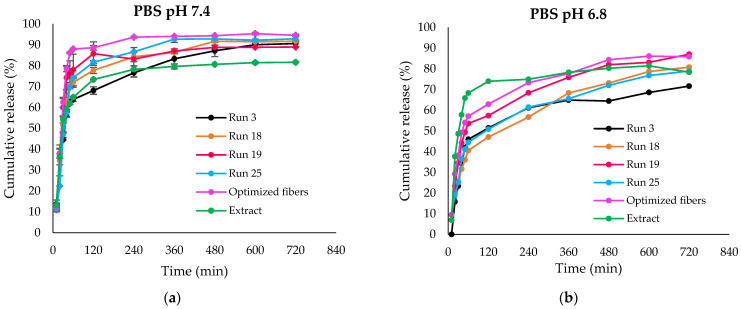
Cumulative rhein release of fiber samples and extract at pH 7.4 (**a**) and pH 6.8 (**b**).

**Figure 9 polymers-16-00183-f009:**
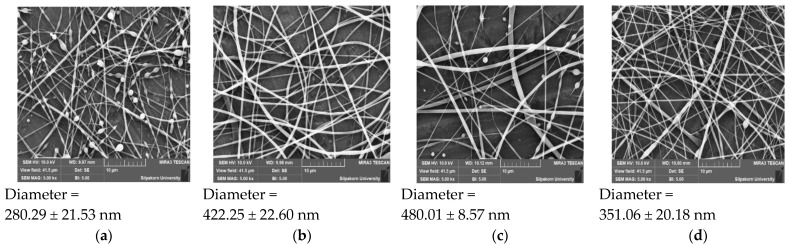
SEM images showing the effect of individual factors on fiber diameter. (**a**) Low shellac content (35% *w*/*w*) with 4% *w*/*w* extract content at 16.5 kV and 0.8 mL/h; (**b**) high shellac content (45% *w*/*w*) with 4% *w*/*w* extract content at 16.5 kV and 0.8 mL/h; (**c**) low voltage (9 kV) with shellac-to-extract ratio (40:4) at 0.8 mL/h; and (**d**) high voltage (24 kV) with shellac-to-extract ratio (40:4) at 0.8 mL/h.

**Figure 10 polymers-16-00183-f010:**
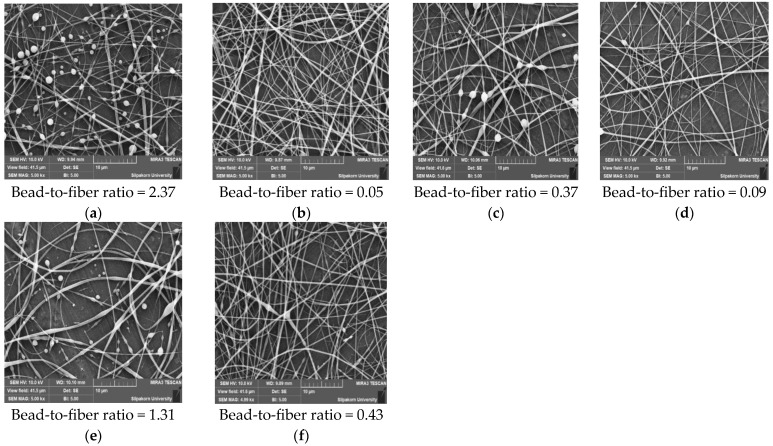
SEM images showing the effect of individual factors on the bead-to-fiber ratio. (**a**) Low shellac content (35% *w*/*w*) with 1% *w*/*w* extract content at 16.5 kV and 0.8 mL/h; (**b**) high shellac content (45% *w*/*w*) with 1% *w/w* extract content at 16.5 kV and 0.8 mL/h; (**c**) low extract content (1% *w*/*w*) with 40% *w*/*w* shellac content at 16.5 kV and 0.4 mL/h; (**d**) high extract content (4% *w*/*w*) with 40% *w/w* shellac content at 16.5 kV and 0.4 mL/h; (**e**) low voltage (9 kV) with shellac-to-extract ratio (40:2.5) at 0.4 mL/h; and (**f**) high voltage (24 kV) with shellac-to-extract ratio (40:2.5) at 0.4 mL/h.

**Figure 11 polymers-16-00183-f011:**
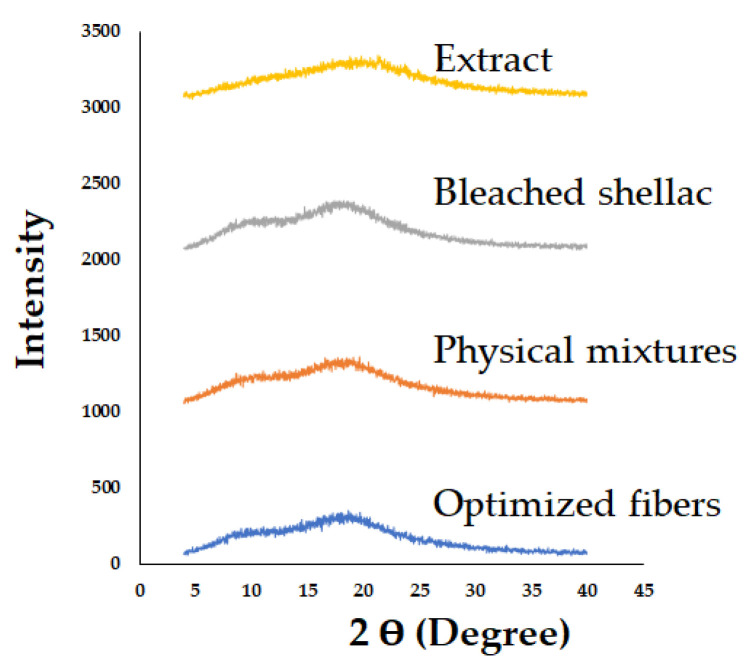
PXRD patterns of bleached shellac, SA leaf extract, their physical mixtures, and optimized electrospun shellac fibers loaded with extract.

**Figure 12 polymers-16-00183-f012:**
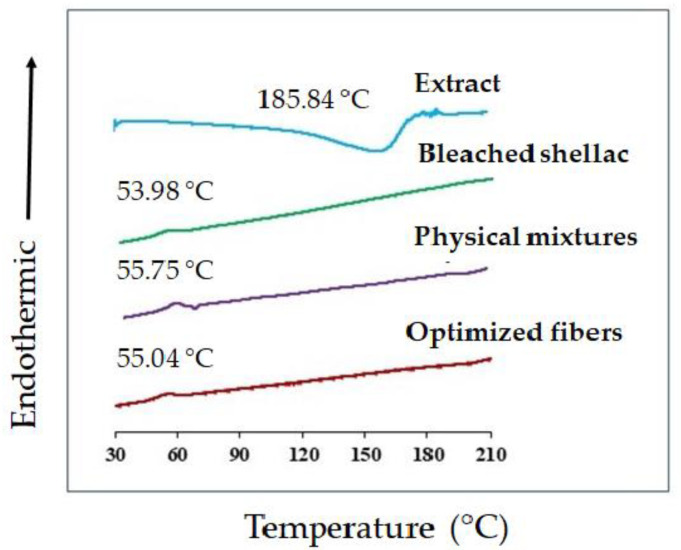
DSC thermograms of bleached shellac, SA leaf extract, their physical mixtures, and optimized electrospun shellac fibers loaded with extract.

**Figure 13 polymers-16-00183-f013:**
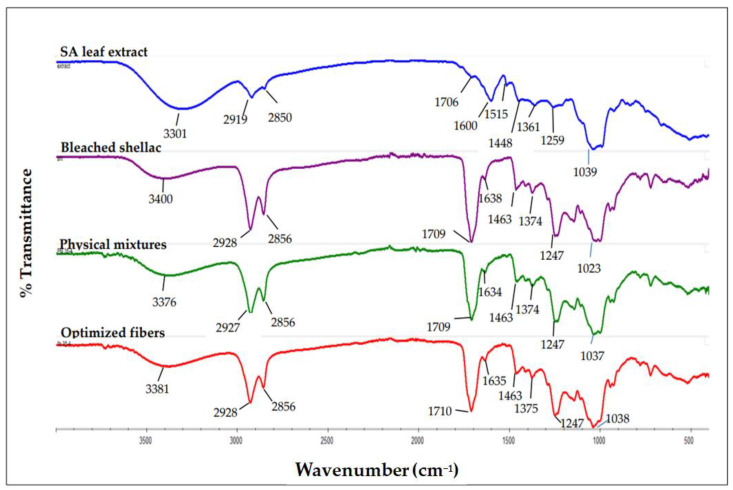
FTIR spectrum of SA leaf extract, bleached shellac, their physical mixtures, and optimized electrospun shellac fibers loaded with extract.

**Figure 14 polymers-16-00183-f014:**
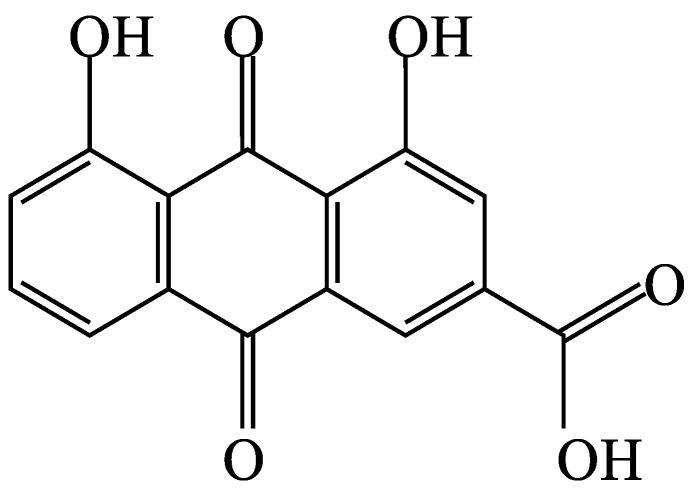
Chemical structure of rhein.

**Figure 15 polymers-16-00183-f015:**
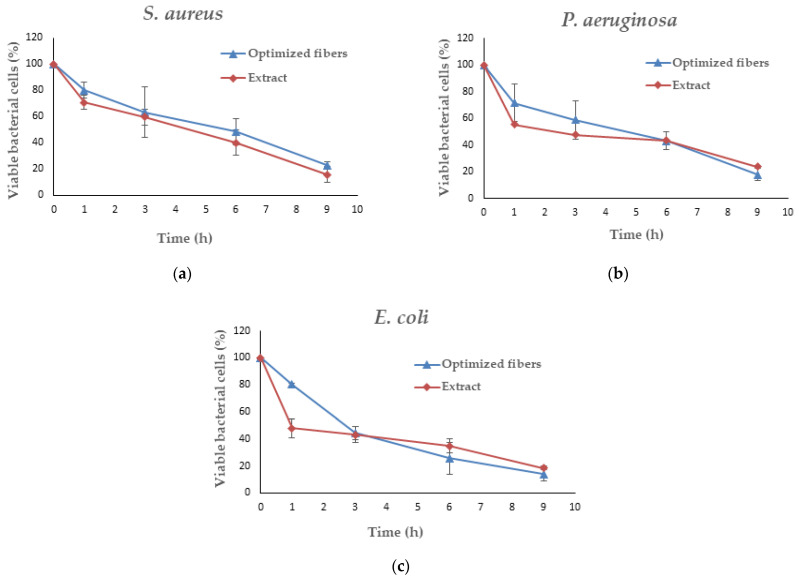
Time–kill kinetics of optimized nanofibers and SA extract against *S. aureus* (**a**), *P. aeruginosa* (**b**), and *E. coli* (**c**).

**Figure 16 polymers-16-00183-f016:**
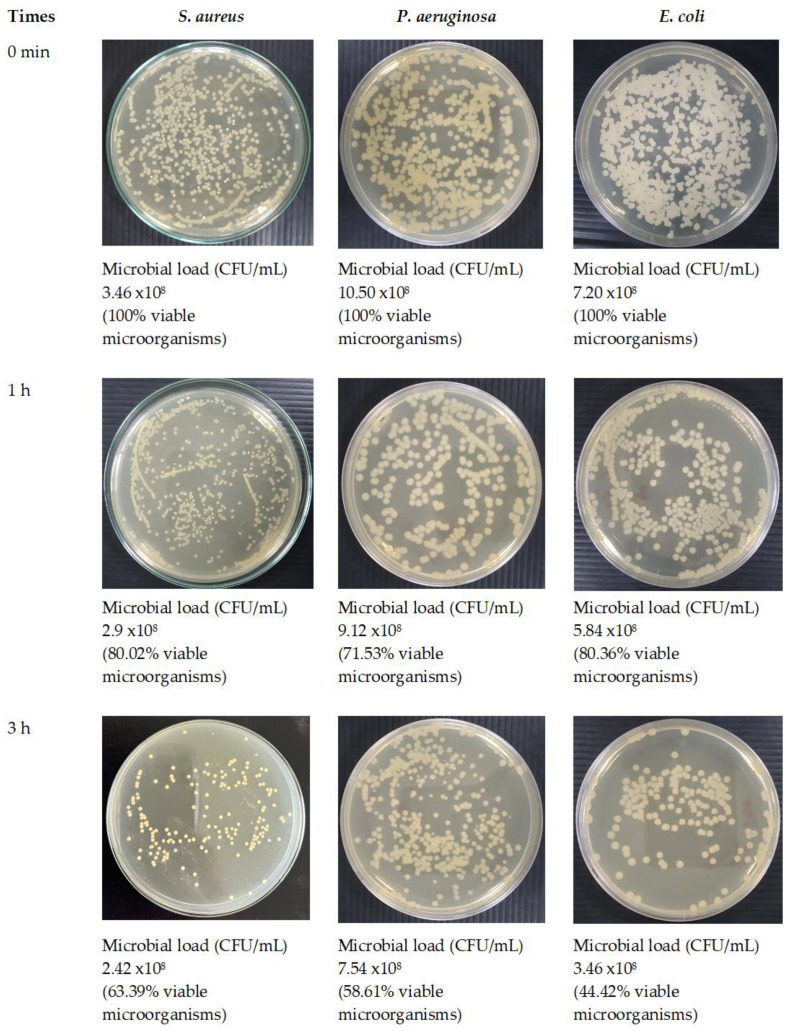
Antimicrobial activities of optimized fibers.

**Table 1 polymers-16-00183-t001:** Experimental ranges and levels of independent variables in FFD.

Independent Variables	Symbol	Level		
		Low (−1)	Center (0)	High (+1)
Shellac content (% *w*/*w*)	X_1_	35.00	40.00	45.00
SA leaf extract content (% *w*/*w*)	X_2_	1.00	2.50	4.00
Applied voltage (kV)	X_3_	9.00	18.00	27.00
Feed rate (mL/h)	X_4_	0.40	0.80	1.20
Needle tip diameter (mm)	X_5_	0.61	0.84	1.06

**Table 2 polymers-16-00183-t002:** Experimental ranges and levels of independent variables in BBD.

Independent Variables	Symbol	Level		
		Low (−1)	Center (0)	High (+1)
Shellac content (% *w*/*w*)	X_1_	35.00	40.00	45.00
SA leaf extract content (% *w*/*w*)	X_2_	1.00	2.50	4.00
Applied voltage (kV)	X_3_	9.00	16.50	24.00
Feed rate (mL/h)	X_4_	0.40	0.80	1.20

**Table 3 polymers-16-00183-t003:** Design matrix and experimental responses of FFD.

Run	Independent Variables					Responses	
	X_1_ Shellac Content (% *w*/*w*)	X_2_ Extract Content (% *w*/*w*)	X_3_ Applied Voltage (kV)	X_4_ Feed Rate (mL/h)	X_5_ Needle Tip Diameter (mm)	R_1_ Fiber Diameter (nm)	R_2_ Bead-To-Fiber Ratio
1	−1 (35.00)	−1 (1.00)	1 (27.00)	1 (1.20)	1 (1.06)	401.02 ± 22.79	4.45 ± 1.30
2	−1 (35.00)	−1 (1.00)	−1 (9.00)	1 (1.20)	−1 (0.61)	398.99 ± 3.71	5.27 ± 1.07
3	0 (40.00)	0 (2.50)	0 (18.00)	0 (0.80)	0 (0.84)	360.93 ± 13.26	0.32 ± 0.11
4	1 (45.00)	−1 (1.00)	1 (27.00)	1 (1.20)	−1 (0.61)	773.52 ± 25.61	0.17 ± 0.02
5	1 (45.00)	1 (4.00)	1 (27.00)	1 (1.20)	1 (1.06)	749.96 ± 15.79	0.16 ± 0.12
6	−1 (35.00)	1 (4.00)	1 (27.00)	−1 (0.40)	1 (1.06)	396.22 ± 10.62	0.56 ± 0.29
7	−1 (35.00)	−1 (1.00)	1 (27.00)	−1 (0.40)	−1 (0.61)	368.51 ± 8.74	1.39 ± 0.61
8	−1 (35.00)	1 (4.00)	−1 (9.00)	−1 (0.40)	−1 (0.61)	374.57 ± 9.27	1.43 ± 0.81
9	1 (45.00)	1 (4.00)	−1 (9.00)	1 (1.20)	−1 (0.61)	394.49 ± 9.52	0.02 ± 0.01
10	1 (45.00)	−1 (1.00)	1 (27.00)	−1 (0.40)	1 (1.06)	727.32 ± 4.43	0.17 ± 0.07
11	−1 (35.00)	1 (4.00)	1 (27.00)	1 (1.20)	−1 (0.61)	330.31 ± 10.38	4.12 ± 0.68
12	1 (45.00)	1 (4.00)	1 (27.00)	−1 (0.40)	−1 (0.61)	680.65 ± 8.98	0.28 ± 0.12
13	−1 (35.00)	1 (4.00)	−1 (9.00)	1 (1.20)	1 (1.06)	343.44 ± 6.93	4.45 ± 1.47
14	1 (45.00)	1 (4.00)	−1 (9.00)	−1 (0.40)	1 (1.06)	419.49 ± 3.24	0.02 ± 0.01
15	1 (45.00)	−1 (1.00)	−1 (9.00)	−1 (0.40)	−1 (0.61)	425.40 ± 11.30	0.02 ± 0.01
16	0 (40.00)	0 (2.50)	0 (18.00)	0 (0.80)	0 (0.84)	323.91 ± 7.84	0.34 ± 0.08
17	−1 (35.00)	−1 (1.00)	−1 (9.00)	−1 (0.40)	1 (1.06)	387.38 ± 2.71	1.03 ± 0.26
18	1 (45.00)	−1 (1.00)	−1 (9.00)	1 (1.20)	1 (1.06)	435.88 ± 5.41	0.02 ± 0.01
19	0 (40.00)	0 (2.50)	0 (18.00)	0 (0.80)	0 (0.84)	349.50 ± 3.19	0.57 ± 0.14

**Table 4 polymers-16-00183-t004:** ANOVA for the selected factorial model.

Source	Sum of Squares	df	Mean Square	F Value	*p*-Value Prob > F
Response 1—Fiber diameter					
Model	359,000	3	120,000	32.97	<0.0001 *
X_1_—Shellac content	161,000	1	161,000	44.49	<0.0001 *
X_3_—Applied voltage	97,320.85	1	97,320.85	26.85	0.0001 *
X_1_X_3_	99,939.92	1	99,939.92	27.58	<0.0001 *
Residual	54,362.18	15	3624.15		
Lack of Fit	53,643.40	13	4126.42	11.48	0.0829 **
Pure Error	718.78	2	359.39		
Cor Total	413,000	18			
R^2^ = 0.8683 Adj R^2^ = 0.8420 Pred R^2^ = 0.8274					
Response 2—Bead-to-fiber ratio					
Model	53.90	3	17.97	64.62	<0.0001 *
X_1_—Shellac content	29.81	1	29.81	107.22	<0.0001 *
X_4_—Feed rate	11.83	1	11.83	42.56	<0.0001 *
X_1_X_4_	12.25	1	12.25	44.06	<0.0001 *
Residual	4.17	15	0.28		
Lack of Fit	4.13	13	0.32	16.47	0.0587 **
Pure Error	0.04	2	0.02		
Cor Total	58.07	18			
R^2^ = 0.9282 Adj R^2^ = 0.9138 Pred R^2^ = 0.9016					

Significant * *p* < 0.01; not significant **.

**Table 5 polymers-16-00183-t005:** Actual (coded) values for the BBD process variables.

Run	Independent Variables				Responses		
	X_1_ Shellac Content (% *w*/*w*)	X_2_ Extract Content (% *w*/*w*)	X_3_ Applied Voltage (kV)	X_4_ Feed Rate (mL/h)	R_1_ Fiber Diameter (nm)	R_2_ Bead-to-Fiber Ratio	R_3_ Entrapment Efficiency (%)
1	0 (40.00)	0 (2.50)	0 (16.50)	0 (0.80)	343.99 ± 13.90	0.33 ± 0.02	82.33
2	0 (40.00)	0 (2.50)	1 (24.00)	−1 (0.40)	335.26 ± 3.92	0.43 ± 0.05	96.04
3	1 (45.00)	0 (2.50)	1 (24.00)	0 (0.80)	446.89 ± 13.01	0.09 ± 0.08	77.44
4	−1(35.00)	−1 (1.00)	0 (16.50)	0 (0.80)	310.52 ± 29.01	2.37 ± 0.49	96.81
5	0 (40.00)	0 (2.50)	0 (16.50)	0 (0.80)	359.51 ± 5.93	0.46 ± 0.12	82.94
6	1 (45.00)	−1 (1.00)	0 (16.50)	0 (0.80)	355.37 ± 10.94	0.05 ± 0.04	70.53
7	0 (40.00)	0 (2.50)	1 (24.00)	1 (1.20)	336.23 ± 20.90	0.16 ± 0.03	90.05
8	−1(35.00)	0 (2.50)	−1 (9.00)	0 (0.80)	325.04 ± 26.16	4.05 ± 0.23	102.99
9	0 (40.00)	1 (4.00)	1 (24.00)	0 (0.80)	351.06 ± 20.18	0.09 ± 0.01	95.52
10	1 (45.00)	0 (2.50)	0 (16.50)	1 (1.20)	414.27 ± 19.82	0.02 ± 0.01	68.97
11	−1(35.00)	0 (2.50)	1 (24.00)	0 (0.80)	312.57 ± 8.51	1.47 ± 0.20	105.42
12	−1(35.00)	1 (4.00)	0 (16.50)	0 (0.80)	280.29 ± 21.53	1.99 ± 0.65	77.55
13	1 (45.00)	1 (4.00)	0 (16.50)	0 (0.80)	422.24 ± 22.60	0.02 ± 0.01	66.39
14	0 (40.00)	0 (2.50)	0 (16.50)	0 (0.80)	319.79 ± 17.05	0.57 ± 0.11	83.54
15	0 (40.00)	1 (4.00)	0 (16.50)	−1 (0.40)	276.04 ± 10.89	0.09 ± 0.04	76.70
16	1 (45.00)	0 (2.50)	0 (16.50)	−1 (0.40)	390.48 ± 15.29	0.03 ± 0.01	58.88
17	−1(35.00)	0 (2.50)	0 (16.50)	−1 (0.40)	259.94 ± 23.12	2.13 ± 0.13	99.81
18	0 (40.00)	0 (2.50)	−1 (9.00)	1 (1.20)	426.59 ± 11.20	1.65 ± 0.44	80.84
19	0 (40.00)	1 (4.00)	0 (16.50)	1 (1.20)	277.07 ± 10.24	0.07 ± 0.02	87.74
20	0 (40.00)	−1 (1.00)	0 (16.50)	1 (1.20)	291.10 ± 7.12	0.34 ± 0.06	93.81
21	0 (40.00)	0 (2.50)	0 (16.50)	0 (0.80)	310.52 ± 2.16	0.39 ± 0.07	90.24
22	0 (40.00)	−1 (1.00)	1 (24.00)	0 (0.80)	356.75 ± 23.34	0.22 ± 0.03	99.44
23	0 (40.00)	1 (4.00)	−1 (9.00)	0 (0.80)	480.01 ± 8.57	1.16 ± 0.40	77.87
24	1 (45.00)	0 (2.50)	−1 (9.00)	0 (0.80)	523.86 ± 6.69	0.25 ± 0.06	59.84
25	−1(35.00)	0 (2.50)	0 (16.50)	1 (1.20)	226.20 ± 5.42	2.62 ± 0.97	88.39
26	0 (40.00)	0 (2.50)	0 (16.50)	0 (0.80)	318.21 ± 2.27	0.38 ± 0.10	82.19
27	0 (40.00)	0 (2.50)	−1 (9.00)	−1 (0.40)	462.28 ± 7.76	1.31 ± 0.47	74.99
28	0 (40.00)	−1 (1.00)	0 (16.50)	−1 (0.40)	332.74 ± 10.42	0.37 ± 0.12	85.82
29	0 (40.00)	−1 (1.00)	−1 (9.00)	0 (0.80)	410.33 ± 6.35	1.48 ± 0.69	95.13

**Table 6 polymers-16-00183-t006:** ANOVA for the quadratic model reducing the response surface.

Source	Sum of Squares	df	Mean Square	F Value	*p*-Value Prob > F
Response 1—Fiber diameter					
Model	118,000	3	39,162.78	43.96	<0.0001 *
X_1_—Shellac content	58,597.40	1	58,597.40	65.77	<0.0001 *
X_3_—Voltage	19,954.60	1	19,954.60	22.40	<0.0001 *
X_3_^2^	38,936.35	1	38,936.35	43.70	<0.0001 *
Residual	22,273.41	25	890.94		
Lack of Fit	20,585.04	21	980.24	2.32	0.2151 ***
Pure Error	1688.37	4	422.09		
Cor Total	139,800	28			
R^2^ = 0.8406 Adj R^2^ = 0.8215 Pred R^2^ = 0.7835					
Response 2—Bead-to-fiber ratio					
Model	27.59	6	4.60	162.82	<0.0001 *
X_1_—Shellac content	16.73	1	16.73	592.40	<0.0001 *
X_2_—Extract content	0.17	1	0.17	5.87	0.0241 **
X_3_—Voltage	4.61	1	4.61	163.31	<0.0001 *
X_1_X_3_	1.46	1	1.46	51.84	<0.0001 *
X_1_^2^	3.94	1	3.94	139.49	<0.0001 *
X_3_^2^	1.18	1	1.18	41.94	<0.0001 *
Residual	0.62	22	0.03		
Lack of Fit	0.59	18	0.03	3.78	0.1032 ***
Pure Error	0.03	4	8.63 × 10^−3^		
Cor Total	28.21	28			
R^2^ = 0.9780 Adj R^2^ = 0.9720 Pred R^2^ = 0.9546					
Response 3—Entrapment efficiency					
Model	3537.04	5	707.41	21.82	<0.0001 *
X_1_—Shellac content	2377.95	1	2377.95	73.34	<0.0001 *
X_2_—Extract content	297.57	1	297.57	9.18	0.0060 *
X_3_—Voltage	434.97	1	434.97	13.42	0.0013 *
X_1_^2^	169.60	1	169.60	5.23	0.0317 **
X_3_^2^	198.54	1	198.54	6.12	0.0211 **
Residual	745.75	23	32.42		
Lack of Fit	699.71	19	36.83	3.20	0.1340 ***
Pure Error	46.04	4	11.51		
Cor Total	4282.80	28			
R^2^ = 0.8259 Adj R^2^ = 0.7880 Pred R^2^ = 0.7052					

Significant * *p* < 0.01; ** 0.01 < *p* < 0.05; not significant ***.

**Table 7 polymers-16-00183-t007:** Criteria for the optimization of electrospinning conditions.

Name	Goal	Lower Limit	Upper Limit
X_1_: Shellac content (% *w*/*w*)	in range	35.00	45.00
X_2_: Extract content (% *w*/*w*)	in range	1.00	4.00
X_3_: Applied voltage (kV)	in range	9.00	24.00
X_4_: Feed rate (mL/h)	equal to 0.8 mL/h	0.40	1.20
Fiber diameter (nm)	in range	250	500
Bead-to-fiber ratio	in range	0.02	0.30
Entrapment efficiency (%)	in range	95.00	105.00

**Table 8 polymers-16-00183-t008:** Optimum conditions with desirable results.

No	Applied Voltage (kV)	Feed Rate (mL/h)	Shellac Content (% *w*/*w*)	Extract Content (% *w*/*w*)	Fiber Diameter (nm)	Bead-to-Fiber Ratio	Entrapment Efficiency (%)	Desirability
1	23.99	0.80	38.50	3.80	335.36	0.30	95.07	1.00

**Table 9 polymers-16-00183-t009:** Confirmation.

Response	Prediction	Experimental Value	95% PI Low	95% PI High
Fiber diameter	335.36	305.77 ± 10.93	291.59	379.13
Bead-to-fiber ratio	0.30	0.29 ± 0.05	0.02	0.57
Entrapment efficiency	95.07	96.38 ± 1.42	86.11	104.03

**Table 10 polymers-16-00183-t010:** Solution properties of different shellac–extract ratios.

Shellac Content (% *w*/*w*)	SA leaf Extract Content (% *w*/*w*)	Viscosity (mPa · S) (Mean ± SD)	Conductivity (µS) (Mean ± SD)	Surface Tension (mN/m) (Mean ± SD)
35.00	0.00	30.81 ± 1.74	94.17 ± 0.15	28.15 ± 0.15
40.00	0.00	59.49 ± 0.98	77.27 ± 0.31	28.56 ± 0.23
45.00	0.00	132.85 ± 1.22	71.50 ± 1.39	28.94 ± 0.06
35.00	1.00	35.89 ± 0.61	170.80 ± 3.36	27.64 ± 0.31
35.00	2.50	48.18 ± 2.82	178.50 ± 0.10	28.52 ± 0.08
35.00	4.00	52.71 ± 1.29	187.80 ± 1.01	30.23 ± 0.28
40.00	1.00	73.57 ± 1.22	160.10 ± 0.10	28.24 ± 0.16
40.00	2.50	91.89 ± 2.09	165.47 ± 1.29	28.64 ± 0.07
40.00	4.00	108.96 ± 4.36	179.17 ± 0.74	29.08 ± 0.05
45.00	1.00	159.41 ± 3.81	158.00 ± 0.56	30.60 ± 0.11
45.00	2.50	183.61 ± 1.42	160.40 ± 0.10	30.32 ± 0.09
45.00	4.00	267.98 ± 6.94	176.50 ± 1.18	32.05 ± 0.19
38.50 (Optimized)	3.80 (Optimized)	88.36 ± 3.15	180.20 ± 0.66	29.11 ± 0.09

## Data Availability

Upon request, the corresponding author will provide access to the data utilized to substantiate the conclusions drawn in this study.

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
