# Peer review of "Fabrication and Optimization of Electrospun Shellac Fibers Loaded with Senna alata Leaf Extract"

_polymers, 2024, doi:10.3390/polym16020183_

Round 1

Reviewer 1 Report

Comments and Suggestions for Authors

This manuscript is a good attempt to fabricate the electrospun shellac fibers with leaf extract. The authors provided sufficient data to support their claims and conclusions. To be honest, this is a very good manuscript. However, there are still some typos and blurry font in the figures. Therefore, it is suggested to accept this work after a minor revision. The comments and suggestions about this work are described as follows:

1. The font size in the tables in Figure 1 is too small. It is suggested to enlarge the font size in Figure 1.

2. In Figure 3, the authors claimed that their data values were normally distributed. However, a very obvious deviation can be observed in Figure 3b. The authors should explain this discrepancy.

3. The authors provided tons of experimental values, which is very good! Accordingly, it is suggested to unify the significant figures in all the tables. 

4. The authors should double-check all the tables with correct values. We still can find several typos like R2 without superscript in Table 6.

5. The font sizes of tick labels in Figures 5 and 6 are too small to be read. It is suggested to provide a bigger font size for the figures.

6. The authors should unify the same unit with the same format in the manuscript. For example, %w/w or % w/w?

7. The authors should add the titles of x- and y-axis with units in Figure 7.

8. In Figure 9, the authors provided the sizes of nanofibers. However, from the figures, we can see the diameter difference between different nanofibers. Therefore, it is suggested to provide the average size with the standard deviation.

Comments on the Quality of English Language

The quality of English in this manuscript is very good!

Reviewer 2 Report

Comments and Suggestions for Authors

The authors report fabrication and optimization of electrospun shellac fibers loaded with senna alata Leaf Extract. The shellac content had the greatest impact on both fiber diameter and bead formation. The optimum electrospinning conditions were identified as a voltage of 24 kV, a solution feed rate of 0.8 mL/h, and a shellac-extract ratio of 38.5:3.8. These conditions produced nanosized fibers with a diameter of 306 nm, a low bead-to-fiber ratio 24 of 0.29, and an extract entrapment efficiency of 96% within the fibers. The optimized nanofibers demonstrated antimicrobial efficacy against diverse pathogens. The paper is well written. The results are very interesting. However, some points of the manuscript should be improved. Specific comments are given below.

1.   Many values of R2 < 0.9. Does it mean that the results are not good enough?

2.   The authors should compare with their fibers with other fibers previously reported in the papers.

3.   The authors should analysis the component of Senna alata Leaf Extract.

4.   Cumulative rhein release of fiber samples and extract at pH 7.4 (a) and pH 6.8 (b). The authors should compare the results in one picture.

Comments on the Quality of English Language

Minor editing of English language required
